# Latent Representation Matters: Human-like Sketches in One-shot Drawing Tasks

**Victor Boutin**[1,2,3], **Rishav Mukherji**[3], **Aditya Agrawal**[3], **Sabine Muzellec**[2,3], **Thomas Fel**[1,3], **Thomas Serre**[1,3], **Rufin VanRullen**[1,2]

[1]Artificial and Natural Intelligence Toulouse Institute, Université de Toulouse, Toulouse, France.
[2]Centre de Recherche Cerveau & Cognition CNRS, Universite de Toulouse, France
[3]Carney Institute for Brain Science, Brown University
victor_boutin@brown.edu

## Abstract

Humans can effortlessly draw new categories from a single exemplar, a feat that has long posed a challenge for generative models. However, this gap has started to close with recent advances in diffusion models. This one-shot drawing task requires powerful inductive biases that have not been systematically investigated. Here, we study how different inductive biases shape the latent space of Latent Diffusion Models (LDMs). Along with standard LDM regularizers (KL and vector quantization), we explore supervised regularizations (including classification and prototype-based representation) and contrastive inductive biases (using SimCLR and redundancy reduction objectives). We demonstrate that LDMs with redundancy reduction and prototype-based regularizations produce near-human-like drawings (regarding both samples' recognizability and originality) – better mimicking human perception (as evaluated psychophysically). Overall, our results suggest that the gap between humans and machines in one-shot drawings is almost closed.

## 1 Introduction

For cognitive scientists, human drawings offer a window into the brain, providing tangible insights into its visual and motor internal processes [1]. For instance, drawings have been used in clinical settings to screen for perceptual impairments following brain trauma or Alzheimer's disease [2, 3], to assess perceptual disorders in autistic individuals [4–6] or to investigate perceptual changes during child development [7, 8] (see [1] for a recent review). Drawing tasks have also proven instrumental for exploring how the brain generalizes to novel visual categories [9–11]. Cognitive psychologists routinely use the one-shot drawing task to understand how human observers can reliably form new object categories from just one exemplar [12, 13]. From a computational viewpoint, this task is ill-defined because of the infinite number of possible sets of samples that could be associated with that exemplar. Yet, humans can effortlessly produce drawings that are not only easily recognizable but also original (i.e., sufficiently distinct from the reference exemplar) [12]. This remarkable capability suggests that the brain leverages powerful representational inductive biases – yet to be discovered – to form novel categories.

Computer scientists have started to make progress in identifying some of the inductive biases for machine learning algorithms to learn from limited data. For one-shot classification tasks, a particularly effective representational inductive bias is to design an embedding space where samples of the same category, whether seen during training or not, cluster closely. This approach spans a wide range of models ranging from representations learned via contrastive objective functions [14–16], prototype-based representations [17, 18] or metric matching losses [19, 20]. Conversely, for one-shot generation tasks, researchers have preferred architectural over representational inductive biases. For instance,

38th Conference on Neural Information Processing Systems (NeurIPS 2024).

novel architectures based on Generative Adversarial Networks (GANs) or Variational Auto-Encoders (VAEs) have incorporated forms of spatial attention [21] or contextual integration [22–24]. Recent advances in diffusion models [25, 26] make them particularly promising for one-shot generation tasks. Indeed, clever conditioning on a context vector [24] or directly using guidance from the exemplar [27] has led to powerful one-shot diffusion models [28]. Such a guidance mechanism has also proven successful in Latent Diffusion Models (LDMs) [29], which use a Regularized AutoEncoder (RAE) to compress input data and a diffusion model to learn the RAE's latent distribution. These diffusion models have started to close the gap with humans in the one-shot drawing task [30] (see section 2 for related work on one-shot learning). While better conditioning mechanisms have driven improvements in one-shot generative models, the potential of shaping their input space with representational inductive biases inspired by one-shot classification remains largely unexplored. This raises the question: "Do representational inductive biases from one-shot classification help narrow the gap with humans in one-shot drawing tasks ?"

In this article, we use Latent Diffusion Models (LDMs [29]) to address this question. LDMs combine the flexibility of the Regularized AutoEncoder (RAE), in which one can seamlessly include various representational inductive biases in the latent space via regularization, with the high expressivity of the diffusion model. Herein, we study the impact of 6 different regularizers corresponding to distinct representational inductive biases. They are categorized into 3 groups. The first group, which serves as a baseline, includes the **KL** and the **vector quantization** regularization approaches typically used in LDMs [29]. The second group involves supervised regularizers: a **classification** loss that promotes discriminative features mapping with categorical training labels and a **prototype**-based objective function that clusters samples with their respective prototypes in an embedding space. The third group features contrastive learning regularization schemes with the **SimCLR** and **Barlow** losses. The **SimCLR** objective function keeps a sample and its augmented view close in the embedding space but far apart from other samples' views. In contrast, the **Barlow** loss ensures that features of similar samples are decorrelated from those of dissimilar ones.

We compare those regularized LDMs against humans on the one-shot drawing task. Such a task offers a leveled playfield in which humans and machines can create sketches that are directly comparable using established evaluation frameworks [31, 30, 12] (see section 2 for related work). More specifically, our comparison focuses on two metrics to evaluate the quality of sketches produced by humans and machines – based on how distinct from the exemplar and how recognizable they are [31] – and on the alignment between humans' and machines' perceptual strategies. For the latter, we describe a novel method to generate importance maps highlighting category-diagnostic features in LDMs. These maps are then directly compared against importance maps derived from human observers obtained through psychophysics experiments. Our results show that LDMs using **prototype**-based and redundancy-reduction (with the **Barlow** twin objective) regularization techniques are further closing the gap with humans. These results are supported by both the sample's similarity and the feature importance maps alignment. Overall, our contributions can be summarized as follows:

- We introduce novel representational inductive biases in Latent Diffusion Models. In particular, we draw inspiration from losses that have proven effective in one-shot classification tasks (with the **prototype**-based, **Barlow** and **SimCLR** objective functions) to regularize the latent space of LDMs.

- We derive a novel explainability method to generate LDMs' feature importance maps that highlight category diagnostic features.

- We systematically compare the sketches and feature importance maps derived from humans and machines, and we show that LDMs with **prototype**-based and **Barlow** regularization significantly narrow the gap with humans on the one-shot drawing task.

Our work underscores the critical role of well-designed representational inductive biases in achieving human-like performance in one-shot drawing tasks. It also sets the stage for developing generative models that are better aligned with humans.

## 2    Related work

**Representation learning for one-shot classification tasks:** Learning representations that bring unseen samples (from the query set) close to the exemplars (in the support set) has proven effective in one-shot classification. The historical approach, called metric learning, aims at creating a feature

space in which the distances between the query and support sets are preserved [20, 19, 32, 33]. However, the limited number of samples in the support set restricts these networks' ability to recognize novel classes. This limitation becomes more pronounced in the one-shot setting as the support set contains only one sample (the exemplar). To address this, the field has shifted towards prototype-based representations. Rather than trying to preserve the distances between query and support samples, such networks learn an embedding space in which the query samples cluster near the support samples [17, 34, 35]. Contrastive learning, a self-supervised learning approach, offers another effective solution to mitigate sample scarcity by augmenting the training set. This method learns an embedding space where positive pairs (a sample and its augmented version) are close together, and distant from negative pairs (augmented views from different instances) [14, 15, 36–39]. Among alternative methods, the SimCLR algorithm [14] uses a cosine similarity between samples whereas the Barlow-twins network [15] leverages the correlation matrix between features to dissociate positive and negative pairs. In this article, we use the **prototype**-based [17], the **SimCLR** [14] and the **Barlow** twins [15] objectives to regularize RAEs latent space. For additional mathematical details, see section 4.1 for the prototype-based loss and section A.2.3 for SimCLR and Barlow.

**Generative models for one-shot image generation tasks:** Some of the main techniques involve including information from the support set into the generative process, a method known as conditioning. For instance, the Neural Statistician uses a context vector containing summary statistics from the support sets, which is then concatenated with a VAE latent space [22, 24, 40]. Similarly, GANs leverage a compressed representation of the support set as a conditioning mechanism [23]. Such a mechanism has also been used successfully to either condition [41–43, 29] or guide the denoising process of diffusion models [27, 28] and latent diffusion models [29]. Here, we leverage LDMs with classifier-free guided diffusion models [27]. Such a diffusion process has been shown to well approximate human drawings in one-shot drawing tasks [30].

**Human-machine comparison in one-shot drawing tasks:** Cognitive scientists have developed various methods to compare the generalization abilities of machines and brains on drawing tasks. Lake et al. [44] introduced the Omniglot challenge in which both humans and machines are tasked with drawing symbols from categories represented by a single exemplar (see [45] for a review on the challenge). The authors evaluated the drawings' recognizability in a visual Turing test where humans (or classifiers) had to distinguish between human-drawn and machine-generated symbols [11]. Additional metrics, including classification uncertainty and semantic similarity, were also used to compare drawings produced by humans and machines under different time constraints [46, 8]. While these evaluation frameworks provide useful insights into a sample's recognizability, they do not measure how the diversity of model-generated samples compares to those created by humans. The "originality vs. recognizability" framework [31] mitigates this issue by adding the originality metric. An originality score quantifies the similarity between the original exemplar and its corresponding variations (see section 5.1 for details on this evaluation framework). This evaluation framework has been used to benchmark the generalization performance of mainstream generative models – Diffusion models [47], GANs [48] and VAEs [49] – against humans in the one-shot drawing setting [30]. Although Diffusion models come closest to human performance, a noticeable gap remained in this study. In this article, we use the "originality vs. recognizablility" framework from Boutin et al. [31] to evaluate representational inductive biases in Latent Diffusion Models. In particular, we demonstrate that effective biases in one-shot classification tasks also prove efficient in the one-shot drawing task.

## 3 Datasets

As done in previous work [31, 30, 11], we use the Omniglot [11] and the QuickDraw-FS [30] datasets to compare humans and machines on the one-shot drawing task. These datasets, made of handwritten symbols or drawings, offer a fair environment for comparing the generation abilities of humans and machines [11, 46, 31, 30]. It is important to note that natural images generation is a task beyond human capability, making it unsuitable for a fair comparison between humans and machines.

**Omniglot** contains $1,623$ categories of handwritten characters from $50$ different alphabets, with $20$ samples per class [11]. This article uses a downsampled version of the dataset (size: $48 \times 48$ pixels). We train the models on a training set composed of all available symbols minus 3 symbols per alphabet left aside for the test set (similar to [21]). All the results on the Omniglot dataset are in the Appendix (see A.6).

**QuickDraw-FS** is made from drawings of the *Quick, Draw !* challenge [50]. In this challenge, human participants are asked to produce drawings in less than 20 seconds when presented with an object name. The categories are, therefore, made with semantically consistent samples that do not necessarily represent the same visual concept (e.g., the "phone" object category might contain corded phones, smartphones, phones with rotary dials, etc). The Quickraw-FS dataset mitigates this issue with categories representing the same visual concepts (see A.1 for more details). This dataset is ideally suited for purely visual one-shot generation tasks [30]. It contains 665 categories with 500 samples each. The training set is made of 550 randomly selected categories, and 115 are left aside for the testing set. We downsampled the drawings to $48 \times 48$ pixels to keep computational resources manageable.

For each category in both datasets, we extract a 'prototypical' sample, selected in the center of the category cluster to condition the one-shot generative models (see A.1 for more details on the exemplar selection).

## 4 One-shot Latent Diffusion Models

The one-shot image generation task involves synthesizing variations of a visual concept not seen during training. Let $\mathbf{x} \in \mathbb{R}^D$ denote the image variation and $\mathbf{y} \in \mathbb{R}^D$ the exemplar. Latent Diffusion Models (LDMs) are composed of 2 distinct stages: a first stage leverages a Regularized AutoEncoder (RAE) that extracts a latent representation $\mathbf{z} \in \mathbb{R}^d$ ($d \ll D$) for each image (see green boxes in Fig. 1), and a second stage consisting of a diffusion model that learns the latent distribution (orange boxes in Fig. 1). In the one-shot setting, the diffusion model is conditioned by $\mathbf{z_y}$, the latent representation of $\mathbf{y}$. We call $\mathbf{c}$ the category label of the training set (a one-hot vector).

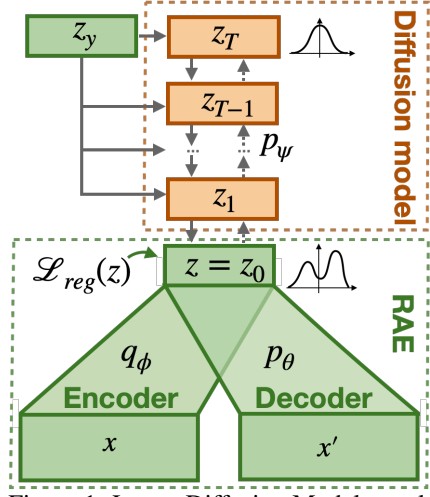

Figure 1: Latent Diffusion Models stack a diffusion model (orange) on top of an Auto-Encoder (green).

### 4.1 Regularized Auto-Encoders (RAEs)

To describe the RAE, we use a probabilistic formulation in which $q_\phi(\mathbf{z}|\mathbf{x})$ is the recognition model (or the encoder), and $p_\theta(\mathbf{x}|\mathbf{z})$ is the decoder. We train the RAEs by minimizing $\mathcal{L}_{RAE}$ (Eq. 1). In this equation, the first term is a reconstruction loss (computed with a $\ell_2$ distance), and the second term ($\mathcal{L}_{reg}$) covers a wide range of regularization losses. $\mathcal{L}_{reg}$ includes the representational inductive biases we study in this article. Those inductive biases fall into 3 groups: the standard LDM regularizers, the supervised regularizers, and the contrastive regularizers.

$$\min_{\theta,\phi} \mathcal{L}_{RAE} \quad \text{s.t.} \quad \mathcal{L}_{RAE} = -\mathbb{E}_{\mathbf{z} \sim q_\phi(.|\mathbf{x})}\left[\log p_\theta(\mathbf{x}|\mathbf{z})\right] + \beta\mathcal{L}_{reg}(\mathbf{z}) \tag{1}$$

**Standard regularizers (KL and VQ):** The **KL divergence** in Eq. 2 forces each coordinate of the latent vector to be distributed following a pre-determined distribution (e.g Gaussian distribution, as in the VAE [49]). The **vector quantized** loss in Eq. 3 transforms the continuous latent code $\mathbf{z}$ into a discrete code $\mathbf{z_q}$ using the nearest entry in a codebook $\mathcal{Z} = \{\mathbf{e_i}\}_{i=1}^{K}$ with the quantization operator: $\mathbf{z_q} = n_\mathcal{Z}(\mathbf{z})$ (s.t. $n_\mathcal{Z} : \mathbf{z} \to \arg\min_{e_i} \|\mathbf{z} - \mathbf{e_i}\|_2$ as in the VQ-VAE [51]). This quantization operation being non-differentiable, backpropagation is achieved using a stop-gradient operation $sg[\cdot]$ to provide a gradient estimator. We provide an extensive mathematical description of the VQ-VAE in App. A.2.1.

$$\mathcal{L}_{KL} = \mathbb{KL}(q_\phi(\mathbf{z}|x)||p(\mathbf{z})) \quad (\text{with } p(\mathbf{z}) = \mathcal{N}(0,\mathbf{I})) \qquad \text{VAE} \tag{2}$$

$$\mathcal{L}_{VQ} = (\|sg[\mathbf{z}] - \mathbf{z_q}\|_2^2 - \|sg[\mathbf{z_q}] - \mathbf{z}\|_2^2) \qquad \text{VQ-VAE} \tag{3}$$

**Supervised regularizers (Classif. and Proto.):** The **classification** regularizer forces discriminative features by minimizing the cross-entropy between the true labels (**c**) and the softmax of the logits. Here the logits are learned by a linear layer ($h_\theta^{CL}$) stacked on the latent space (Eq. 4). While the **classification** loss is supervised by the true categorical labels, the **prototype**-based loss is supervised

by the exemplars themselves (as in the Prototypical Net [17]). The **prototype**-based loss learns a metric space in which classification can be performed by computing distances between the variations and their corresponding exemplars (i.e., the prototypes)(see Eq. 5). Here, the metric space is linked to the latent space of the RAE through a linear layer ($h_\theta^{PR}$). Intuitively, the **prototype**-based loss finds an embedding space where the variations will be close (in terms of $\ell_2$ distance) from their prototypes. See A.2.2 for more details.

$$\mathcal{L}_{CL} = \mathcal{CE}(h_\theta^{CL}(\mathbf{z}), \mathbf{c}) \qquad \qquad \text{Classif.} \qquad (4)$$

$$\mathcal{L}_{PR} = \mathbb{E}_{\mathbf{z_y} \sim \mathbf{q}_\phi(\cdot|\mathbf{y})}\big[ -\log(\mathrm{softmax}(\big\|h_\theta^{PR}(\mathbf{z}) - h_\theta^{PR}(\mathbf{z_y})\big\|_2)]  \qquad \textbf{Proto.} \qquad (5)$$

**Contrastive regularizers (SimCLR and Barlow):**   Contrastive learning algorithms learn representations that are invariant under different distortions (i.e., data augmentations). Here we define two data-augmentation operators, $\tau^A(\cdot)$ and $\tau^B(\cdot)$, that transform the variations $\mathbf{x}$ into $\mathbf{x^A} = \tau^A(\mathbf{x})$ and $\mathbf{x^B} = \tau^B(\mathbf{x})$, respectively. We denote $\mathbf{z^A} = q_\phi(\cdot|\mathbf{x^A})$ and $\mathbf{z^B} = q_\phi(\cdot|\mathbf{x^B})$ the projection of $\mathbf{x^A}$ and $\mathbf{x^B}$ into the RAE latent space, respectively. The **SimCLR** regularizer is based on the InfoNCE loss: it maximizes the similarity between the representation of a sample and its augmented view while minimizing the similarity with negative pairs (augmented views of different instances) [14]. The **Barlow** regularizer (as in the Barlow twins [15]) forces the cross-correlation matrix between $\mathbf{z^A}$ and $\mathbf{z^B}$ to be as close to the identity matrix as possible. This causes the embedding vectors of distorted versions of samples to be similar while minimizing the redundancy between the components of these vectors. Said differently, the **SimCLR** loss shapes the space based on the samples' similarity, while the **Barlow** operates on the correlation between the features of the samples. For conciseness, we have included the mathematical derivations and details on the data augmentation we used in App. A.2.3.

We leverage standard convolutional architectures (from [52]) to parametrize both the encoder and the decoder. The resulting autoencoder has a 1D bottleneck ($d = 128$ for QuickDraw-FS and $d = 64$ for Omniglot). We refer the reader to App. A.3.1 for complete architectural and training details of the RAE. In the rest of the article, we evaluate the impact of these regularizations by exploring the effect of $\beta$ (see Eq. 1) on LDMs.

## 4.2   Diffusion Model

The LDM second stage is a diffusion model that learns the data distribution in the latent space of the RAE. Diffusion models progressively denoise a pure noise $\mathbf{z_T} \sim \mathcal{N}(0, \mathbf{I})$ into a clean latent representation $\mathbf{z_0} := \mathbf{z}$ through a sequence of partially denoised variables $\{\mathbf{z_i}\}_{i=1}^T$. The goal is then to learn a transition probability $p_\psi(\mathbf{z_{t-1}}|\mathbf{z_t})$ that approximates a noise injection operator $\nu_t(.)$ so that $\mathbf{z_t} = \nu_t(\mathbf{z_0}) = \sqrt{\bar{\alpha}_t}\mathbf{z_0} + \sqrt{1 - \bar{\alpha}_t}\epsilon$ ($\bar{\alpha}_t$ is an hyperparameter of the diffusion schedule, and $\epsilon$ a Gaussian noise). The Denoising Diffusion Probabilistic Model (DDPM) [47] reduces the learning of $p_\psi(\mathbf{z_{t-1}}|\mathbf{z_t})$ to the optimization of a simple autoencoder $\epsilon_\psi$ trained to predict the noise from a degraded latent representation $\mathbf{z}_t$ (see A.4 for mathematical justification):

$$\underset{\psi}{\arg\min} \, \mathbb{E}_{\substack{\mathbf{z_0} \sim q_\phi(\cdot|\mathbf{x}) \\ \mathbf{z_y} \sim q_\phi(\cdot|\mathbf{y})}} \left[ \big\|\epsilon_\psi\big(\nu_t(\mathbf{z_0}), \mathbf{z_y}, t\big) - \epsilon\big\|_2^2 \right] \quad \text{s.t.} \quad \epsilon \sim \mathcal{N}(0, \mathbf{I}) \quad \text{and} \quad t \sim \mathcal{U}(1, T) \quad (6)$$

In Eq. 6, $\mathbf{z_y}$ denotes the latent representation of the exemplar $\mathbf{y}$. Eq. 6 could be interpreted as a denoising score matching objective [53], so the optimal model $\epsilon_{\psi*}$ matches the following score function:

$$\nabla_{\mathbf{z}_t} \log p_{\psi^\star}(\mathbf{z}_t|\mathbf{z_y}) \approx -\frac{1}{\sqrt{1 - \bar{\alpha}_t}} \epsilon_{\psi^\star}(\mathbf{z}_t, \mathbf{z_y}) \qquad (7)$$

The autoencoder-like model $\epsilon_\psi(., \mathbf{z_y}, t)$ is a 1D Unet conditioned on the time variable $t$ and $\mathbf{z_y}$ (see A.4.3 for details on the architecture and the training of the Unet). Herein, we use a classifier-free guided version of the DDPM [27] with the following score function:

$$\nabla_{\mathbf{z}_t} \log p_{\psi^\star, \gamma}(\mathbf{z}_t|\mathbf{z_y}) = (1 + \gamma)\nabla_{\mathbf{z}_t} \log p_{\psi^\star}(\mathbf{z}_t|\mathbf{z_y}) - \gamma\nabla_{\mathbf{z}_t} \log p_{\psi^\star}(\mathbf{z}_t) \qquad (8)$$

This formulation introduces a guidance scale $\gamma$ (we use $\gamma$=1) to tune how much the conditioning signal influences the final score. Such a formulation has shown effective in one-shot settings [28, 30]. Note that each term on the RHS of Eq. 8 is computed with the same network $\epsilon_\psi$ using Eq. 7. $\epsilon_\psi$ is simply conditioned on a non-informative signal to compute $\log p_{\psi^\star}(\mathbf{z}_t)$. We remind the reader that the training of the diffusion model begins only after the RAE training is complete, and occurs exactly

identically, regardless of the type of regularization used. The quality of images generated by the diffusion model thus directly serves to compare the different regularizations. The code to train all described models is available at `http://anonymous.4open.science/r/LatentMatters-526B`.

## 5 Results

### 5.1 Originality vs. Recognizabilty

To compare humans and machines in the one-shot drawing task, we first use the originality vs. recognizability framework [31, 30]. This framework leverages 2 critic networks to evaluate the samples produced during the testing phase. The recognizability is quantified using the classification accuracy of a one-shot classifier [17], while the originality is measured using the average distance between the variations and their corresponding exemplars. This distance is computed in the feature space of a self-supervised model [14]. Importantly, both human-drawn and machine-generated samples are evaluated using the same 2 critic networks. This ensures that any potential biases in the critic networks are minimized, leading to a more balanced comparative analysis. Note that the originality is normalized across all tested models to range between $0$ and $1$. Here, we use the same originality vs. recognizability framework setting as that used in Boutin et al. [30]. Importantly, the originality vs. recognizability plots should be interpreted based on how close the models are to the human data point (grey star in Fig. 3), rather than focusing solely on their individual originality or recognizability scores. In simple terms, a model that effectively mimics human drawings should fall near the human data point. Note also that there is an inherent trade-off between originality and recognizability: while recognizability assesses how likely the data point falls within the classifier decision boundary, originality measures how 'diffuse' the sample distribution is. Therefore a very original agent (producing highly diverse samples) will tend to have a low recognizability as the samples are likely to fall outside of the classifier decision boundary.

In Fig. 3, we first evaluate how increasing the regularization weights (i.e. the $\beta$ in Eq. 1) for each regularizer (taken separately) affects the similarity of LDM samples to human drawings. To do so, we report the originality and the recognizability values for LDM samples trained with different $\beta$ values (see data points in Fig. 3). We use a parametric fit (least curve fitting methods [54]) to illustrate how increasing $\beta$ affects these scores (see A.5 for more details on the parametric fit computations). We observe a similar concave shape for all curves. As $\beta$ starts increasing, the recognizability improves while the originality decreases (except for **VQ** regularizer). Beyond a certain $\beta$ value, the recognizability declines, and the originality increases. In particular, the maximum recognizability values for **KL** and **VQ** (obtained with $\beta_{KL} = 10^{-5}$ and $\beta_{VQ} = 5$) match those of a diffusion model trained in the pixel space and barely exceed those of a non-regularized LDM (see Fig. 3a). Increasing the weight of the **prototype**-based regularizer substantially reduces the distance to human compared to the **classification** regularizer (the minimal distance to human is $0.04$ for $\beta_{PR} = 5 \cdot 10^2$ vs. $0.15$ for $\beta_{CL} = 5$, see Fig. 3b). Among the contrastive regularizers, **Barlow** regularization significantly reduces the distance to human compared to the **SimCLR** one (the minimal distance to human is $0.08$ with $\beta_{BAR} = 30$ vs. $0.12$ with $\beta_{SimCLR} = 10^{-2}$, see Fig. 3c). A visual inspection of the samples tends to corroborate these results (see Fig. 2 and A.7 for more samples). We observe similar trends for all tested regularizers on the Omniglot dataset (see A.6).

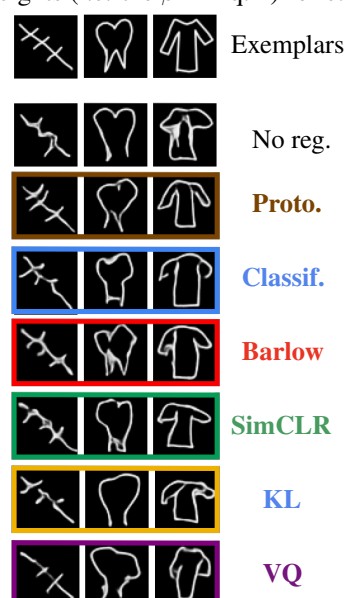

Figure 2: **Samples from LDMs w/ different regularizers.** The LDMs correspond to the larger data points in Fig. 3.

Overall, our findings indicate that not all regularizers are created equal. For supervised regularizers (see Fig. 3b), the **prototype**-based regularizer generates more recognizable samples compared to the **classification** regularizer. This is expected since the classifier focuses on separating categories in the training set, which may not be ideal for unseen categories in the one-shot setting [19, 17]. In contrast, the **prototype**-based regularizer clusters samples near their prototypes, leading to less

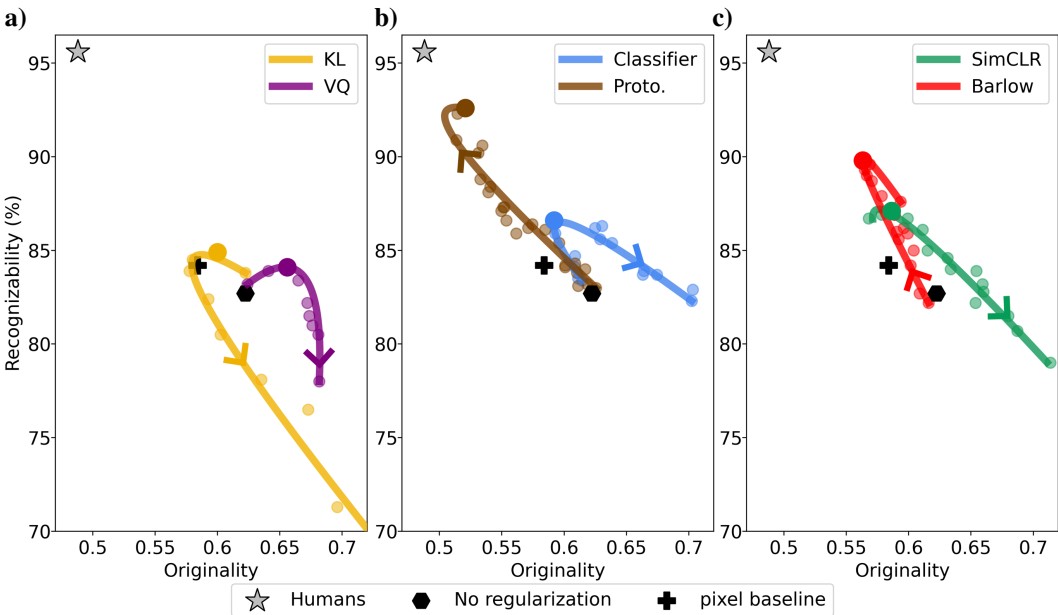

Figure 3: **Effect of increasing the regularization weights on the originality vs recognizability framework (QuickDraw-FS dataset).** Each data point represents an LDM trained with different values of regularization weights ($\beta$). The curves represent the parametric fits, oriented in the direction of an increase of $\beta$. **a):** For the LDMs with "standard" regularizers, the $\beta$ is applied on the **KL** ($\mathcal{L}_{KL}$ in Eq. 2) or on the **VQ** regularizers ($\mathcal{L}_{VQ}$ in Eq. 3). **b):** For the supervised regularizers, the $\beta$ is applied on the **CL** ($\mathcal{L}_{CL}$ in Eq. 4) or on the **prototype**-based regularizers ($\mathcal{L}_{PR}$ in Eq. 5). **c):** For the contrastive regularizers, the $\beta$ is applied on the **SimCLR** ($\mathcal{L}_{SimCLR}$ in Eq. 14) or on the **Barlow** regularizers ($\mathcal{L}_{Bar}$ in Eq. 15). See A.5 for more information on the range of $\beta$ we have explored for each regularizer. Larger data points indicate models whose performance is closer to that of humans for each type of regularization. For comparison, we include an LDM leveraging a non-regularized RAE (hexagon marker) and a diffusion model trained directly on the pixel space (cross marker). The human performance corresponds to the recognizability and originality computed on human drawings (shown with a grey star).

overfitting and better transferability, which is valuable for few-shot tasks [55]. Our experiments confirm that the **prototype**-based regularizer generalizes better for one-shot drawing. In Fig. 3c, the **Barlow** regularization outperforms the **SimCLR** regularizer in recognizability, likely due to Barlow's effective feature disentangling [15]. These features transfer well to new datasets, making Barlow more suitable for the one-shot drawing task. Overall, our results demonstrate that effective representational inductive biases in few-shot learning also enhance performance in one-shot drawing.

We now study the effect of the regularizers when they are used in combination. In particular, we have systematically explored the following combinations of regularizers **Barlow** + **Prototype** (Fig. 4a), **SimCLR.** + **Prototype** (Fig. 4b), **KL** + **Prototype** (Fig. 4c), **VQ** + **Prototype** (Fig. 4d). We observe that the **Barlow** + **Prototype** and the **KL** + **Prototype** combinations produced the most human-like samples. Those regularizer's combinations are particularly as in both cases the combined recognizability is significantly higher compared to using each regularizer alone. This suggests that clustering samples around their prototypes (using the **Prototype** regularizer) within a disentangled space (achieved via the **KL** or **Barlow** regularizer) enhances generalization. In contrast, the **VQ** + **Prototype** and the **SimCLR** + **Prototype** combinations show little to no improvements.

## 5.2   Comparing humans and LDM perceptual strategies

While the originality vs. recognizability framework allows us to compare human and machine performances in the one-shot drawing task, it does not reveal the strategies each uses to generalize to new categories. To address this, we aim to compare the visual strategies more directly via feature importance maps. These maps emphasize the most salient features to recognize a drawing.

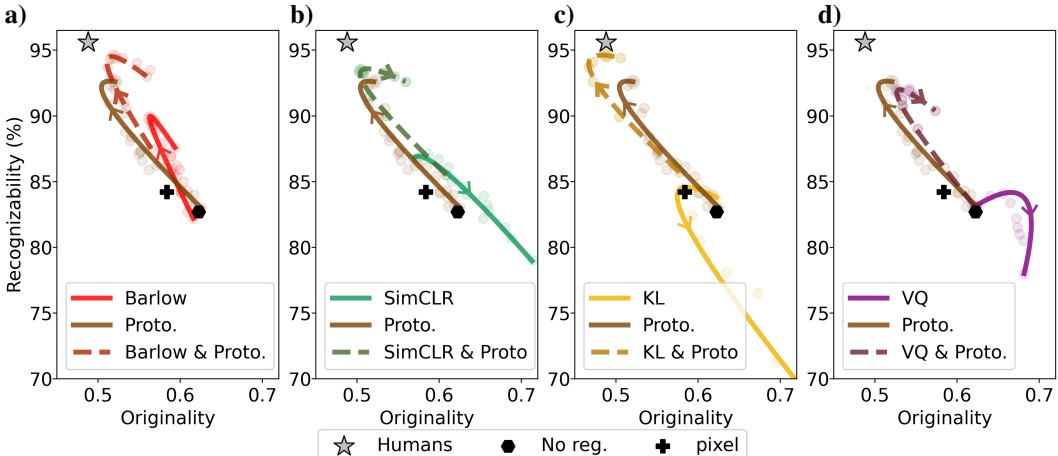

Figure 4: **Combined effect of the regularization weights on the originality vs recognizability framework (QuickDraw-FS dataset).** Each data point represents an LDM trained with a combination of 2 different regularizers. All combinations include the **prototype**-based regularizers. The curves represent the parametric fits, oriented in the direction of an increase of $\beta$. **a): Barlow** and **prototype**-based regularizers applied either separately (plain lines) or in combination (dashed-line). When applied in combinations, only the weight of the **prototype**-based regularizer is modified (with $\beta = 30$ for **Barlow**). **b): SimCLR** and **prototype**-based regularizers. When applied in combinations, only the weight of the **prototype**-based regularizer is modified, the **SimCLR** is set to $\beta = 1$. **c): KL** and **prototype**-based regularizers. When applied in combinations, only the weight of the **prototype**-based regularizer is modified, the **KL** is set to $\beta = 1e - 3$. **d): VQ** and **prototype**-based regularizers. When applied in combinations, only the weight of the **prototype**-based regularizer is modified, the **VQ** is set to $\beta = 20$. See caption in Fig. 3.

Previous research has demonstrated that by summing the absolute values of the diffusion scores ($\nabla_{z_t} \log p_\psi(z_t|z_y)$) throughout all diffusion steps, one can create heatmaps that highlight salient features in a diffusion model's generation process [30]. Here, we adapt this heuristic to make it compatible with LDMs. This involves projecting each intermediate noisy state ($\mathbf{z_t}$) back to pixel space using the RAE's decoder ($p_\theta(\cdot|\mathbf{z_t})$). To do so, we use the chain rule, and we multiply each diffusion score by the Jacobian of the RAE decoder w.r.t $\mathbf{x_t}$ (denoted $J_{\log p_\theta(\cdot|\mathbf{z_t})}(\mathbf{x_t})$). For each variation $\mathbf{x}$ and its corresponding exemplar $\mathbf{y}$, we can therefore compute a heatmap using Eq. 9 (see A.8.1 for mathematical details). Then, we average 10 of these heatmaps, obtained with the same exemplar but for different variations belonging to the same category. This process allows us to mitigate intra-class variations while focusing on category-specific features. We call this average the feature importance map (see A.8.2 to visualize feature importance maps).

$$\phi(\mathbf{x}, \mathbf{y}) = \sum_{i=0}^{T} \left| J_{\log p_\theta(\cdot|\mathbf{z_t})}(\mathbf{x_t}) \nabla_{\mathbf{z_t}} \log p_\psi(\mathbf{z_t}|\mathbf{z_y}) \right| \text{ with } \mathbf{z_y} \sim q_\phi(\cdot|\mathbf{y}) \tag{9}$$

We derived human feature importance maps using psychophysical data from Boutin et al. [30] (data shared by the original authors). The authors collected human saliency maps through an online psychophysics experiment based on a similar protocol to the ClickMe experiment [56]. In this experiment, participants were presented with drawings and were asked to draw on regions important for categorization (see App. S in [30] for more details on the experimental protocol). We averaged the heatmaps across participants and drawings within the same category to obtain the feature importance maps we compared with those of machines (see A.8.3 for visualizing feature importance maps).

In Fig 5, we compare humans and LDMs feature importance maps. For each regularizer, we select the LDMs that produce the most human-like sketches (highlighted with larger data points in Fig. 3). Note that we exclude the **VQ**-regularized LDM from this analysis because it produces irrelevant feature importance maps, possibly due to the non-differentiability of the quantization process (see Fig. A.15). In Fig. 5a, we showcase examples of the obtained feature importance maps for all other LDMs' regularizations (see also A.8.2) and for humans (see also A.8.3). We qualitatively observe that the LDMs regularized with the **Barlow** and the **prototype**-based objectives tend to focus on sparse features. This particular aspect seems to be shared with the human feature importance maps. We compute the Spearman rank correlation to quantify the similarity between human and machine

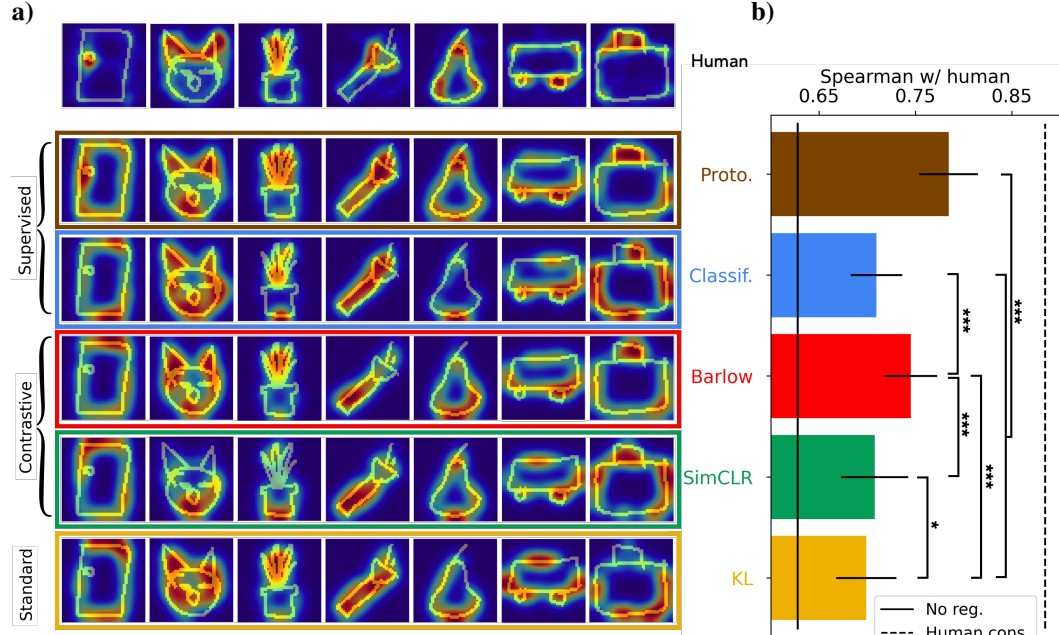

Figure 5: **Feature importance maps comparison. a)** The visualizations include feature importance maps for humans (top row) and LDMs (six bottom rows). All the maps are overlaid on exemplars. Hot vs. cold pixels show image locations that are more vs. less important. Maps for humans were computed using psychophysical data from Boutin et al. [30]. For the LDMs, they are obtained for each category by averaging $\phi(\mathbf{x}, \mathbf{y})$ (see Eq. 9) over 10 different image variations ($\mathbf{x}$) belonging to the same category. The models' maps are computed on the more human-like LDMs for each regularization (larger data points in Fig. 5). **b)** Spearman's rank correlation coefficient between humans and LDMs feature importance maps. The error bar is computed as the standard deviation of the Spearman coefficients over all categories (25 in total). Stars indicate the p-value ($\star\star\star : p < 10^{-3}$ and $\star : p < 5.10^{-2}$) of pair-wise statistical test between models (Wilcoxon signed-rank test, see A.8.4). The black line corresponds to an LDM without any regularization. The dashed line is the human consistency (0.88), it quantifies how much two populations of humans agree with each other on feature importance maps (see A.8.3 for details on the human consistency computation).

feature importance maps (see Fig. 5b). To make sure that the correlation comparison between the different LDMs is significant, we have computed pairwise statistical tests (Wilcoxon signed-rank test, see A.8.4). Our results show that all considered regularizations correlate significantly more with human feature importance maps than non-regularized LDMs. In addition, the **prototype**-based regularizer produces the feature importance maps with the highest correlation with humans and is significantly above all other tested regularizations ($p < 10^{-3}$). In the human-alignment ranking, the **Barlow**-regularized LDM follows the **prototype**-based LDM, also showing a significantly higher Spearman correlation coefficient than **KL**, **classification**, **SimCLR** regularizers ($p < 10^{-3}$). All other pair-wise statistical tests (between **KL**, **classification**, **SimCLR**) are not significant enough to draw a meaningful ranking.

## 6  Conclusion

In this article, we used Latent Diffusion Models (LDMs) to study the effect of representational inductive biases for one-shot drawing tasks. We explore 6 different regularizers: **KL**, **vector quantization**, **classification**, **prototype**-based, **SimCLR** and **Barlow** regularizers. We analyzed the human/LDMs alignment from two (independent) perspectives: their performance relative to humans on the one-shot drawing task (with the recognizability vs. originality framework in section 5.1) and the similarity of the underlying visual strategies (with the feature importance maps in 5.2). Overall, we observe a clear alignment between the 2 analyses on the following points:

- All regularized LDMs have an optimal regularization weight ($\beta$) where they are more aligned with humans than their non-regularized counterparts.

- The **prototype**-based regularizer is showing the best matches with human performance and attentional strategy.
- In the one-shot drawing tasks, the samples' human-likeness could be further improved by combining the **prototype**-based regularizer with either the **KL** or the **Barlow** regularizers.

In conclusion, we observe that all representational inductive biases "are not created equal". However, some of them (**prototype**-based and **Barlow** regularizers) do narrow the gap with humans in the one-shot drawing task.

# 7   Limitations

In this article, we tested six representational inductive biases, a small number considering the extensive range available in the representation-learning literature. This field encompasses hundreds of inductive biases that have proven successful in one-shot classification tasks. Therefore, other representational inductive biases might align better with human performance, both in terms of sample similarity and visual strategy. Our goal wasn't to test all possible biases but to demonstrate that some of them can significantly narrow the gap with humans in one-shot drawing tasks.

Another limitation of this article lies in the recognizability vs. originality framework we are using to evaluate the drawings. This framework leverages 2 critic networks to evaluate the sample's originality and recognizability. There's no guarantee these networks align with human perceptual judgments. Thus, the recognizability and originality scores might not reflect human perception accurately. However, since both human and model outputs are evaluated using the same pre-trained critic networks, the comparison remains fair.

# 8   Discussion

It is noteworthy that the **KL** and **VQ** regularizers, commonly used to train LDMs on natural images (as in StableDiffusion [29]) are not the best-performing regularizers in the one-shot drawing task. Our study indicates that the **prototype**-based and the **Barlow** regularizers, not tested yet on LDMs trained on natural images, hold a significant potential for enhancing their one-shot ability. From a single image of a new vehicle prototype or of a new fashion item design, a generative model trained with these regularizers could produce relevant variations – an ability that current commercial applications still struggle with (see Fig. A.8.5).

Interestingly, the 2 inductive biases that align most closely with humans are directly related to prominent neuroscience theories. The **prototype**-based objectives provide an instantiation of the prototype theory of recognition and memory [57–61], suggesting that humans use prototype similarity to recognize novel objects. Similarly, the **Barlow** regularization is inspired by Barlow's redundancy reduction theory [62, 63], which posits that the brain encodes statistically independent features to eliminate redundancy (and minimize energy consumption). The effectiveness of these regularizations provides hints that the brain may use similar inductive biases to generalize to new categories. In terms of brain inspiration, although we use LDMs to model humans' one-shot generation abilities, we do not claim that these neural networks constitute a realistic model of brain processes. It is indeed unlikely that humans generate samples by iteratively denoising random noise. More biologically plausible generative models might further help to obtain better models of human behavior (e.g., see [64–68]).

With this paper, we highlight how specific representational inductive biases, included in the input space of generative models, can help bridge the gap with human capabilities. We believe these biases will allow advanced models to generalize and create as effectively as humans do, leading to exciting advancements in technology and creativity.

## Aknowledgement

This work was funded by the European Union (ERC, GLoW, 101096017), ANITI (Artificial and Natural Intelligence Toulouse Institute) and the French National Research Agency (ANR-19-PI3A-0004). Additional funding was provided by ONR (N00014-24-1-2026) and NSF (IIS-1912280, IIS-2402875 and EAR-1925481). Computing hardware supported by NIH Office of the Director grant S10OD025181 via the Center for Computation and Visualization (CCV).

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

# A Appendix/Supplementary Information

## A.1 QuickDraw-FS dataset

The QuickDraw-FS dataset is built from the samples of the *Quick, Draw !* challenge [50]. In this online experiment (`https://quickdraw.withgoogle.com`), participants have to draw an object when presented with the category name. The resulting dataset is made of 345 object categories, with approximately $150,000$ drawings per category. The experimental protocol of the *Quick, Draw !* challenge forces the participants to produce drawings that are semantically related to the category name, but those drawings do not necessarily represent the same visual concepts. For example, the "alarm clock" category includes digital and analogic types of alarm clocks, which represent 2 different visual concepts (see Fig.A.1). This property makes the original *Quick, Draw !* dataset not optimal for purely visual one-shot generation tasks.

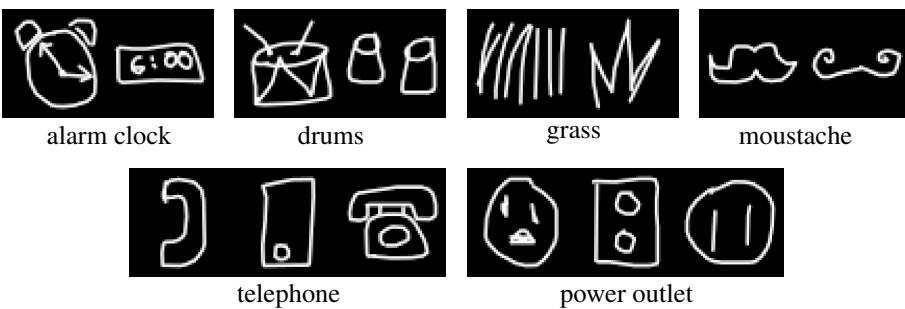

alarm clock      drums      grass      moustache

telephone      power outlet

Figure A.1: Examples of distinct visual concepts belonging to the same object category in the *Quick, Draw !* dataset.

To mitigate this issue, previous work has proposed the QuickDraw-FS dataset. In this dataset, new categories are formed based on the visual similarity of the drawings (see Appendix A in [30]). The authors have used clustering techniques in the latent space of the contrastive learning algorithms to compute the infer the new categories. The resulting dataset is made of categories representing one single visual concept. Using this dataset, one can extract a "prototype" exemplar – at the center of the cluster – to exemplify the category visual concepts. We include examples of drawing variations and their corresponding exemplars in Fig. A.2.

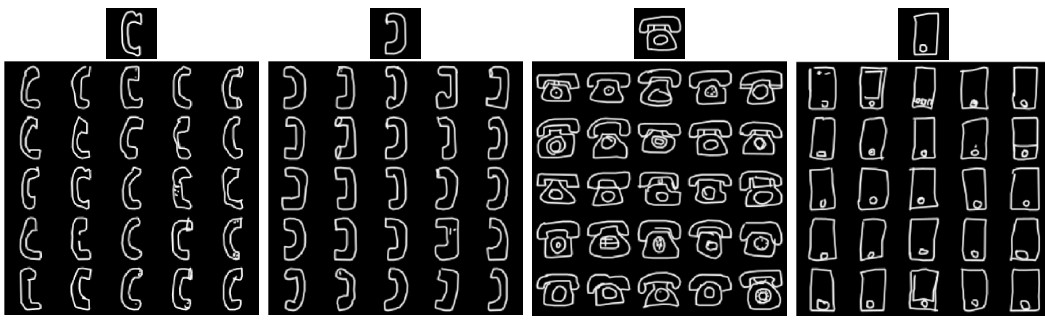

Figure A.2: Illustration of the samples and the corresponding exemplars for 4 categories of the QuickDraw-FS dataset. The small image located on the top represents the exemplars of the different visual concepts. The $5 \times 5$ grid of drawings represents the corresponding visual concepts (randomly sampled in the cluster.

## A.2 Regularized AutoEncoders

### A.2.1 VQ-VAE

Let us define a codebook $\mathcal{Z} = \{\mathbf{e}_i\}_{i=1}^K$ made of $K$ elements (also called codewords). Each codeword has a dimension s : $\mathbf{e}_i \in \mathbb{R}^s$. The Vector-Quantized Variational AutoEncoder (VQ-VAE) [51] can be decomposed into 3 stages: i) an encoder $q_\phi(\mathbf{z}|\mathbf{x})$ mapping the input data $\mathbf{x}$ to a continuous latent vector $z \in \mathbb{R}^d$, ii) a discretizing operator denoted $n_\mathcal{Z}(z)$ which transforms $\mathbf{z}$ into a discretized latent vector $\mathbf{z}_q$, and iii) a decoder $p_\theta(\mathbf{x}|\mathbf{z}_q)$ mapping $\mathbf{z}_q$ to a reconstructed image $\mathbf{x}$. The discrete latent code $\mathbf{z}_q$ is calculated using a nearest-neighbor look-up in the codebook $\mathcal{Z}$ (see Eq. 10). Said differently, each element of the continuous latent vector $\mathbf{z_i}$ is replaced by the nearest $\mathbf{e_j}$ in the codebook (here the $i$ index corresponds to the $i$-th coordinate of $\mathbf{z}$):

$$\mathbf{z_{q}}_i = n_\mathcal{Z}(\mathbf{z_i}) = \arg\min_{\mathbf{e_j} \in \mathcal{Z}} \|\mathbf{z}_i - \mathbf{e}_j\| \tag{10}$$

$\mathbf{z_q}$ could then be transformed into a discretized vector by mapping each codeword with its corresponding address in the codebook ($\mathbf{e_j} \to j$). Note that this quantization process is equivalent to defining a posterior distribution following a $K$-way categorical distribution [51].

To learn the resulting networks, one naive way would be to minimize the following loss function :

$$\arg\min_{\phi, \theta, \mathcal{Z}} \mathcal{L}_{VQVAE} \quad \text{with} \quad \mathcal{L}_{VQVAE} = -\mathbb{E}_{\mathbf{z_q} \sim n_\mathcal{Z}(q_\phi(.|\mathbf{x}))}\left[\log p_\theta(\mathbf{x}|\mathbf{z_q})\right] \tag{11}$$

Eq. 11 is a reconstruction loss in which the information first flows through the quantized encoder, (i.e. $n_\mathcal{Z}(q_\phi(.|\mathbf{x}))$), to then produce a reconstructed image (i.e. $\log(p_\theta(\mathbf{x}|\mathbf{z}))$.

However, Eq. 11 cannot be directly optimized as it has no real gradient (the $\arg\min$ function is not derivable). To minimize this loss function, the gradient is then approximated using a straight-through estimator [36]. The straight-through estimator involves copying the gradients from the decoder input to the encoder output. We refer the reader to line 5 in Algo. 1 for practical implementation of the straight-through gradient estimator. Intuitively, since $\mathbf{z}$ is supposed to be very close to $\mathbf{z_q}$, the gradient contains meaningful information for how the encoder has to change to minimize the reconstruction loss. During inference, the nearest embedding $\mathbf{z_q}$ is computed using Eq. 10 and then fed to the decoder. Due to the straight-through operation, the codebook $\mathcal{Z}$ does not receive any gradient information from the reconstruction term. Therefore, the codebook is learned with the simplest dictionary learning algorithm that involves minimizing the $\ell_2$ distance between the quantized vector $\mathbf{z_q}$ and the continuous one $\mathbf{z}$ (i.e. $\|\mathbf{z} - \mathbf{z_q}\|_2^2$). This quantity cannot be directly minimized because there is no gradient flowing from $\mathbf{z_q}$ to $\mathbf{z}$. To mitigate this issue, it is replaced with the estimator term $\|sg[\mathbf{z_q}] - \mathbf{z}\|_2^2 + \|\mathbf{z_q} - sg[\mathbf{z}]\|_2^2$. The full VQ-VAE loss is described in Eq. 12: :

$$\mathcal{L}_{VQVAE} = -\mathbb{E}_{\mathbf{z_q} \sim n_\mathcal{Z}(q_\phi(.|\mathbf{x}))}\left[\log p_\theta(\mathbf{x}|\mathbf{z_q})\right] + \beta_{VQ}(\|sg[\mathbf{z_q}] - \mathbf{z}\|_2^2 + \|\mathbf{z_q} - sg[\mathbf{z}]\|_2^2) \tag{12}$$

The following pseudo-code illustrates how the VQ-VAE is usually implemented (see Algo. 1). We follow a similar implementation:

---

**Algorithm 1: VQVAE pseudo-code**

**Input:** dataset $\mathcal{D}$, model parameters $\pi = (\theta, \phi, \mathcal{Z})$

1 **for** $\mathbf{x}$ in $\mathcal{D}$ **do**
2 $\quad$ $\mathbf{z} = q_\phi(\mathbf{z}|\mathbf{x})$ $\quad$ # encode
3 $\quad$ $\mathbf{z_q} = n_\mathcal{Z}(\mathbf{z})$ $\quad$ # quantize
4 $\quad$ $\mathcal{L}_{reg} = \|sg[\mathbf{z_q}] - \mathbf{z}\|_2^2 + \|\mathbf{z_q} - sg[\mathbf{z}]\|_2^2$ $\quad$ # Quantization loss, $sg[.]$ = stop gradient
5 $\quad$ $\mathbf{z_q} = \mathbf{z} + sg[\mathbf{z_q} - \mathbf{z}]$ $\quad$ # straight-through gradient estimator
6 $\quad$ $\tilde{\mathbf{x}} = p_\theta(\mathbf{x}|\mathbf{z_q})$ $\quad$ # decode
7 $\quad$ $\mathcal{L} = \|\mathbf{x} - \tilde{\mathbf{x}}\|_2^2 + \mathcal{L}_{reg}$
8 $\quad$ $\pi \leftarrow \dfrac{\partial \mathcal{L}}{\partial \pi}$

---

### A.2.2 Prototype-based regularization

Prototypical networks focus on learning an embedding space where data points cluster around a single prototype representation for each class. A prototype is originally defined as the mean vector of

the embedded support points belonging to its class [17]. In the one-shot setting, the support set is reduced to one single sample. Therefore here the prototype and the exemplar are the same.

To achieve the desired embedding space for the autoencoder we regularize the reconstruction loss with a **protoype**-based loss. The loss uses the pairwise $\ell_2$ distance between samples and prototype to derive a probability distribution:

$$\mathcal{L}_{PR} = \mathbb{E}_{\mathbf{z_y} \sim \mathbf{q}_\phi(.|\mathbf{y})} \left[ -\log(\text{softmax}(\left\| h_\theta^{PR}(\mathbf{z}) - h_\theta^{PR}(\mathbf{z_y}) \right\|_2)) \right] \tag{13}$$

In Eq. 13, $h_\theta^{PR}(\mathbf{z_y})$ represents the projection of the prototype in the embedding space while $h_\theta^{PR}(\mathbf{z})$ represents the projections of the sample. See Algo. 2 for more details on the exact implementation of the prototype-based regularized RAE.

---

**Algorithm 2: Prototype-based regularizer pseudo-code**

**Input:** dataset $\mathcal{D} = (\mathbf{x}, \mathbf{y})$, model parameters $\pi = (\theta, \phi)$   # x: variations and y: exemplars
1 **for** *(x, y)* in $\mathcal{D}$ **do**
2     $\mathbf{z} = q_\phi(\mathbf{z}|\mathbf{x})$   # encode variations
3     $\mathbf{z_y} = q_\phi(\mathbf{z_y}|\mathbf{x_y})$   # encode exemplar
4     $d = \left\| h_\theta^{PR}(\mathbf{z}) - h_\theta^{PR}(\mathbf{z_y}) \right\|_2$   # pair-wise distance beteen projected z and $\mathbf{z_y}$
5     $\mathcal{L}_{PR} = -\log(\text{softmax}(d))$
6     $\tilde{\mathbf{x}} = p_\theta(\mathbf{x}|\mathbf{z})$   # decode
7     $\mathcal{L} = \|\mathbf{x} - \tilde{\mathbf{x}}\|_2^2 + \mathcal{L}_{PR}$
8     $\pi \leftarrow \dfrac{\partial \mathcal{L}}{\pi}$

---

### A.2.3   Constrastive regularizers

**Maths and Algorithms:**   Contrastive learning algorithms learn representations that are invariant under different distortions (i.e. data augmentations). Here we use two data-augmentation operators, $\tau^A(\cdot)$ and $\tau^B(\cdot)$, that transform the variations $\mathbf{x}$ into $\mathbf{x^A} = \tau^A(\mathbf{x})$ and $\mathbf{x^B} = \tau^B(\mathbf{x})$, respectively. We denote $\mathbf{z^A}$ and $\mathbf{z^B}$ the latent space projection of $\mathbf{x^A}$ and $\mathbf{x^B}$, respectively (i.e. $q_\phi(\mathbf{z^A}|\mathbf{x^A})$ and $q_\phi(\mathbf{z^B}|\mathbf{x^B})$). Here, we use two different types of contrastive regularizations that are $\mathcal{L}_{SimCLR}$ (see Eq. 14) and $\mathcal{L}_{Bar}$ (see Eq. 15)

$$\mathcal{L}_{SimCLR}(\mathbf{z^A}, \mathbf{z^B}) = \mathbb{E}_{\mathbf{z^A}, \mathbf{z^B}} \left[ -\sum_b \text{sim}(h_\theta^I(\mathbf{z_b^A}), h_\theta^I(\mathbf{z_b^B}))_i + \right.$$
$$\left. \sum_b \log \left( \sum_{b' \neq b} \exp(\text{sim}(h_\theta^I(\mathbf{z_b^A}), h_\theta^I(\mathbf{z_{b'}^B}))_i) \right) \right] \tag{14}$$

$$\mathcal{L}_{Bar}(\mathbf{z^A}, \mathbf{z^B}) = \mathbb{E}_{\mathbf{z^A}, \mathbf{z^B}} \left[ \sum_i \left( 1 - \text{sim}(h_\theta^B(\mathbf{z_{.,i}^A}), h_\theta^B(\mathbf{z_{.,i}^B}))_b \right)^2 + \right.$$
$$\left. \lambda \sum_i \sum_{j \neq i} \left( \text{sim}(h_\theta^B(\mathbf{z_{.,i}^A}), h_\theta^B(\mathbf{z_{.,j}^B}))_b \right)^2 \right] \tag{15}$$

$$\text{with} \, \text{sim}(\mathbf{x}, \mathbf{y})_i = \frac{\langle \mathbf{x}, \mathbf{y} \rangle_i}{\|\mathbf{x}\|_2 \|\mathbf{y}\|_2} \tag{16}$$

In these equations, $b$ indexes the sample in a batch, $i$ indexes the vector component of the embeddings, $h_\theta^I(\mathbf{z})$ and $h_\theta^B(\mathbf{z})$ are linear probe stacked on the RAE latent space. In the Barlow regularizer, we use $\lambda = 5 \times 10^{-3}$. For both networks, the linear probe projects in a space of size 128.

This is important to observe that the scalar product in Eq. 14 is computed along the vector component dimension whereas this is computed along the batch dimension in Eq. 15. Said differently, in Eq. 14 sim computes a square matrix of size (batch size, batch size) (this is a pair-wise similarity matrix between samples) while it is of dimension (feature space dimension, feature space dimension) in Eq. 15 (this is a correlation matrix between vector's coordinate). We refer the reader to Algo. 3 and Algo. 4 for the pseudo-code of the **SimCLR** and the **Barlow** regularizers, respectively.

**Algorithm 3: SimCLR regularizer pseudo-code**

---

**Input:** dataset $\mathcal{D} = \{\mathbf{x}\}$, model parameters $\pi = (\theta, \phi)$   # x: variations

1   **for** *(*$\mathbf{x}, \mathbf{y}$*) in $\mathcal{D}$* **do**
2      $\mathbf{x_A} = \tau^A(\mathbf{x})$   # augment x in $\mathbf{x_A}$
3      $\mathbf{x_B} = \tau^B(\mathbf{x})$   # augment x in $\mathbf{x_B}$
4      $\mathbf{z_A} = q_\phi(\mathbf{z_A}|\mathbf{x_A})$   # encode $\mathbf{x_A}$
5      $\mathbf{z_B} = q_\phi(\mathbf{z_B}|\mathbf{x_B})$   # encode $\mathbf{x_B}$
6      $\mathcal{L}_{reg} = \mathcal{L}_{SimCLR}(\mathbf{z_A}, \mathbf{z_B})$   # see Eq. 14
7      $\tilde{\mathbf{x}} = p_\theta(\mathbf{x}|\mathbf{z_A})$   # decode
8      $\mathcal{L} = \|\mathbf{x} - \tilde{\mathbf{x}}\|_2^2 + \mathcal{L}_{reg}$
9      $\pi \leftarrow \dfrac{\partial \mathcal{L}}{\pi}$

---

**Algorithm 4: Barlow regularizer pseudo-code**

---

**Input:** dataset $\mathcal{D} = \{\mathbf{x}\}$, model parameters $\pi = (\theta, \phi)$   # x: variations

1   **for** *(*$\mathbf{x}, \mathbf{y}$*) in $\mathcal{D}$* **do**
2      $\mathbf{x_A} = \tau^A(\mathbf{x})$   # augment x in $\mathbf{x_A}$
3      $\mathbf{x_B} = \tau^B(\mathbf{x})$   # augment x in $\mathbf{x_B}$
4      $\mathbf{z_A} = q_\phi(\mathbf{z_A}|\mathbf{x_A})$   # encode $\mathbf{x_A}$
5      $\mathbf{z_B} = q_\phi(\mathbf{z_B}|\mathbf{x_B})$   # encode $\mathbf{x_B}$
6      $\mathcal{L}_{reg} = \mathcal{L}_{Bar}(\mathbf{z_A}, \mathbf{z_B})$   # see Eq. 15
7      $\tilde{\mathbf{x}} = p_\theta(\mathbf{x}|\mathbf{z_A})$   # decode
8      $\mathcal{L} = \|\mathbf{x} - \tilde{\mathbf{x}}\|_2^2 + \mathcal{L}_{reg}$
9      $\pi \leftarrow \dfrac{\partial \mathcal{L}}{\pi}$

---

**Augmentations:** The augmentations we use are the same for both regularizers (i.e. $\tau^A(\cdot)$ and $\tau^B(\cdot)$), they are randomly picked among the following transformations:

- **Random resized crop:** with a scale parameter ranging from (0.1, 0.9) and a ratio parameter ranging from (0.8, 1.2). The scale parameter tunes the upper and lower bound of the cropped area, and the ratio parameter defines the lower and upper bound for the aspect of the ratio of the crop.

- **Random affine transformation:** with a rotation parameter varying from ($-15°$ to $15°$), a translation (from $-5$ pixels to $5$ pixels), a zoom (with a ratio from 0.75 to 1.25) and a shearing (from $-10°$ to $10°$)

- **Random perspective transformation:** apply a scale distortion with a certain probability to simulate 3D transformations. The scale distortion we have chosen is 0.5, and it is applied to the image with a probability of $50\%$

### A.3 RAEs training and architectures

#### A.3.1 RAEs architectures

For the encoder, $q_\phi(\mathbf{z}|\mathbf{x})$, and decoder, $p_\theta(\mathbf{x}|\mathbf{z})$, we leverage similar architectures than those proposed in Ghosh et al. [52]. In Table 1 we detail the exact architecture of the RAE encoder and decoder.

| Network | Layer | Input Shape | Output Shape | Param # |
|---|---|---|---|---|
| | Conv2d | [1, 48, 48] | [16, 24, 24] | 256 |
| | BatchNorm2d | [16, 24, 24] | [16, 24, 24] | 32 |
| | ReLU | [16, 24, 24] | [16, 24, 24] | – |
| | Conv2d | [16, 24, 24] | [32, 12, 12] | 8,192 |
| | BatchNorm2d | [32, 12, 12] | [32, 12, 12] | 64 |
| | ReLU | [32, 12, 12] | [32, 12, 12] | – |
| | Conv2d | [32, 12, 12] | [64, 7, 7] | 32,768 |
| Encoder : $q_\phi(\mathbf{z}|\mathbf{x})$ | BatchNorm2d | [64, 7, 7] | [64, 7, 7] | 128 |
| | ReLU | [64, 7, 7] | [64, 7, 7] | – |
| | Conv2d | [64, 7, 7] | [128, 3, 3] | 131,072 |
| | BatchNorm2d | [128, 3, 3] | [128, 3, 3] | 256 |
| | ReLU | [128, 3, 3] | [128, 3, 3] | – |
| | Linear | [128, 3, 3] | [$d$] | 147,584 ($d = 128$) |
| | ConvTranspose2d | [$d$, 1, 1] | [128, 6, 6] | 1,179,648 ($d = 128$) |
| | BatchNorm2d | [128, 6, 6] | [128, 6, 6] | 256 |
| | ReLU | [128, 6, 6] | [128, 6, 6] | – |
| | ConvTranspose2d | [128, 6, 6] | [64, 12, 12] | 131,072 |
| | BatchNorm2d | [64, 12, 12] | [64, 12, 12] | 128 |
| | ReLU | [64, 12, 12] | [64, 12, 12] | – |
| | ConvTranspose2d | [64, 12, 12] | [32, 24, 24] | 32,768 |
| | BatchNorm2d | [32, 24, 24] | [32, 24, 24] | 64 |
| Decoder : $p_\theta(\mathbf{x}|\mathbf{z})$ | ReLU | [32, 24, 24] | [32, 24, 24] | – |
| | ConvTranspose2d | [32, 24, 24] | [16, 48, 48] | 8,192 |
| | BatchNorm2d | [16, 48, 48] | [16, 48, 48] | 32 |
| | ReLU | [16, 48, 48] | [16, 48, 48] | – |
| | ZeroPad2d | [16, 48, 48] | [16, 49, 49] | – |
| | Conv2d | [16, 49, 49] | [1, 48, 48] | 257 |
| | Sigmoid | [1, 48, 48] | [1, 48, 48] | – |

Table 1: The base architecture for all the autoencoders.

Note that for Omniglot and QuickDraw, we have chosen different latent-space sizes (denoted $d$). For Omniglot $d = 64$ and for QuickDraw, $d = 128$.

#### A.3.2 RAEs training details

We train the model using the Mean Squared Error loss with a batch size of 128 for the reconstruction, along with different regularizations to study its effects. For both datasets, we use the Adam optimizer [69] with a weight decay of $10^{-5}$ and a learning rate of $10^{-4}$. The RAEs on the QuickDraw dataset were trained for 200 epochs and 300 epochs on the Omniglot dataset. Note that when trained on the Omniglot dataset, we use a learning rate scheduler in which the learning rate is divided by $4$ every 70 epoch.

## A.4  Latent Diffusion models

In this section, we describe the mathematics behind the latent diffusion models. The following mathematical derivations are mostly derived from Sohl-Dickstein et al. [26], Song and Ermon [25], Ho et al. [47], Rombach et al. [29] and are adapted to match the one-shot generation task and the notations of this paper. Those mathematical derivations are not necessary to understand this article but we include them to make it self-contained.

Herein, we consider a pretrained Regularized AutoEncoder, with an encoder $q_\phi(\mathbf{z}|\mathbf{x})$ and decoder $p_\theta(\mathbf{x}|\mathbf{z})$ that map the input $\mathbf{x} \in \mathbb{R}^D$ to a latent representation $\mathbf{z} \in \mathbb{R}^d$ ($d \ll D$) and inversely, respectively. In the following, we will call indifferently $\mathbf{z}$ or $\mathbf{z_0}$ the latent variable corresponding to the input $\mathbf{x}$. We will also call $\mathbf{z_y}$ the latent variable associated with the exemplar $\mathbf{y}$. The goal of a diffusion model in a one-shot latent diffusion algorithm is to learn the conditional probability of $\mathbf{z_0}$ given the latent representation of the exemplar $\mathbf{z}_y$, we call this probability distribution $p_\psi(\mathbf{z_0}|\mathbf{z}_y)$.

### A.4.1  Diffusion process and noising operator in latent diffusion process

Diffusion models learn the transformation of a pure noise, called $\mathbf{z_T} \in \mathbb{R}^d$, into a fully denoised latent representation $\mathbf{z_0} \in \mathbb{R}^d$. This transformation is progressive, through a sequence of partially denoised latent representations $\{\mathbf{z_i}\}_{i=1}^{T-1} \in \mathbb{R}^{d \times (T-1)}$. In this sequence $\mathbf{z_{t+1}}$ is therefore slightly more noisy than $\mathbf{z_t}$. The idea behind the diffusion model is to learn the transition probability $p_\psi(\mathbf{z_{t-1}}|\mathbf{z_t}, \mathbf{z_y})$. To do so, diffusion models introduce a tractable noising process $r(\mathbf{z_t}|\mathbf{z_{t-1}})$ that gradually injects noise in the latent representation. An illustration of such a directed graphical model is shown in Fig. A.3.

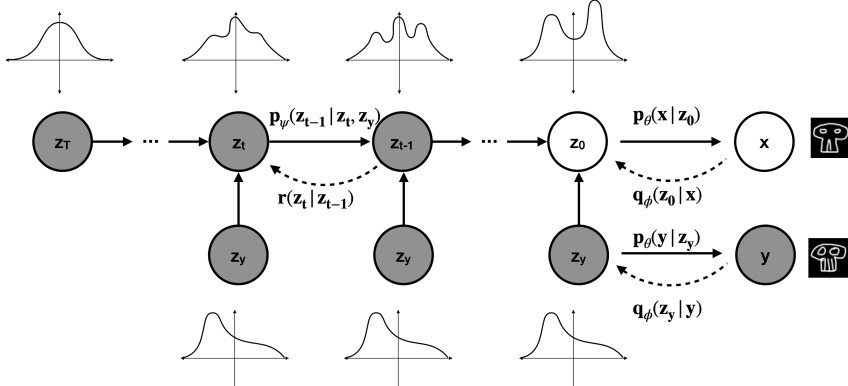

Figure A.3: The directed graphical model considered in this work. Dotted and plain arrows represent the forward (i.e. noise injection) and the reverse processes (i.e. noise removal), respectively. $\mathbf{z_y}$ and $\mathbf{z_0}$ are the latent representations of the exemplar image $\mathbf{y}$ and the image $\mathbf{x}$, respectively (exemplified with skull drawings). $\mathbf{z_i}$ corresponds to the sequence of partially corrupted latent representations. $\mathbf{z_y}$ and $\mathbf{z_0}$ are obtained using the RAE encoder $q_\phi(\mathbf{z}|\mathbf{x})$ and can be mapped to the input space using the RAE decoder $p_\theta(\mathbf{x}|\mathbf{z})$. The 'dummy' distributions located on top of the $\mathbf{z_i}$ variables, illustrate the noise injection process, starting from an 'informative' multimodal distribution to a fully 'uninformative' Gaussian distribution.

Here we describe, in mathematical terms, the noise injection process :

$$r(\mathbf{z}_{1:T}|\mathbf{z_0}) = \prod_{t=1}^{T} r(\mathbf{z}_t|\mathbf{z}_{t-1}) \ \ \text{with} \ \ r(\mathbf{z}_t|\mathbf{z}_{t-1}) = \mathcal{N}(\mathbf{z}_t; \sqrt{1-\beta_t}\mathbf{z}_{t-1}, \beta_t\mathbf{I}) \ \ \text{s.t.} \ \ \{\beta_t \in (0,1)\}_{i=1}^{T} \tag{17}$$

In Eq. 17, $\beta_t$ tunes the step size of the diffusion process. Using the successive product of Gaussian, this process could be reduced to a tractable noising operator $\nu_t(.)$ that injects the right amount of noise at time $t$ to obtain $\mathbf{z_t}$ from $\mathbf{z_0}$:

$$\mathbf{z}_t = \sqrt{\alpha_t}\mathbf{z}_{t-1} + \sqrt{1-\alpha_t}\epsilon \quad \text{with} \quad \epsilon \sim \mathcal{N}(\mathbf{0}, \mathbf{I})$$

$$= \sqrt{\alpha_t \alpha_{t-1}}\mathbf{z}_{t-2}\sqrt{1-\alpha_t\alpha_{t-1}}\epsilon$$

$$= \dots$$

$$= \sqrt{\bar{\alpha}_t}\mathbf{z}_0 + \sqrt{1-\bar{\alpha}_t}\epsilon = \nu_t(\mathbf{z_0}) \quad \text{with} \quad \alpha_t = 1 - \beta_t \quad \text{and} \quad \bar{\alpha}_t = \prod_{i=1}^{t}\alpha_t \tag{18}$$

One could then express the probablity of $\mathbf{z_t}$ given $\mathbf{z_0}$ in a closed form:

$$r(\mathbf{z}_t|\mathbf{z}_0) = \mathcal{N}(\mathbf{z}_t; \sqrt{\bar{\alpha}_t}\mathbf{z}_0, (1-\bar{\alpha}_t)\mathbf{I}) \tag{19}$$

The denoising probabilistic process, recovering the latent representation $\mathbf{z}_0$ from noise, could be parametrized as follows:

$$p_\psi(\mathbf{z}_{0:T}|\mathbf{z_y}) = p_\psi(\mathbf{z}_T|\mathbf{z_y})\prod_{t=1}^{T}p_\psi(\mathbf{z}_{t-1}|\mathbf{z}_t, \mathbf{z_y}) \tag{20}$$

$$\text{with} \quad \begin{cases} p_\psi(\mathbf{z}_{t-1}|\mathbf{z}_t, \mathbf{z_y}) & = \mathcal{N}(\mathbf{z}_t; \mu_\psi(\mathbf{z}_t, t, \mathbf{z_y}), \sigma_t^2\mathbf{I}) \\ p_\psi(\mathbf{z}_T|\mathbf{z_y}) & = s(\mathbf{z}_T) = \mathcal{N}(\mathbf{0}, \mathbf{I}) \end{cases}$$

### A.4.2 Loss of the Denoising Diffusion Probabilistic Model in the Latent Diffusion case

As in VAEs [49], the Evidence Lower Bound of the diffusion model could be recovered using Jensen's inequality [47]:

$$\mathbb{E}_{\mathbf{z}_0\sim r(\mathbf{z}_0)}\log p_\psi(\mathbf{z}_0|\mathbf{z_y}) = \mathbb{E}_{\mathbf{z}_0\sim r(\mathbf{z}_0)}\log\left(\int p_\psi(\mathbf{z}_{0:T}|\mathbf{z_y})d\mathbf{z}_{1:T}\right)$$

$$= \mathbb{E}_{\mathbf{z}_0\sim r(\mathbf{z}_0)}\log\left(\int r(\mathbf{z}_{1:T}|\mathbf{z}_0)\frac{p_\psi(\mathbf{z}_{0:T}|\mathbf{z_y})}{r(\mathbf{z}_{1:T}|\mathbf{z}_0)}d\mathbf{z}_{1:T}\right)$$

$$= \mathbb{E}_{\mathbf{z}_0\sim r(\mathbf{z}_0)}\log\left(\mathbb{E}_{\mathbf{z}_{1:T}\sim r(\mathbf{z}_{1:T}|\mathbf{z}_0)}\left[\frac{p_\psi(\mathbf{z}_{0:T}|\mathbf{z_y})}{r(\mathbf{z}_{1:T}|\mathbf{z}_0)}\right]\right)$$

$$\leq \mathbb{E}_{\mathbf{z}_{0:T}\sim r(\mathbf{z}_{0:T})}\log\left(\frac{p_\psi(\mathbf{z}_{0:T}|\mathbf{z_y})}{r(\mathbf{z}_{1:T}|\mathbf{z}_0)}\right) = -L_{VLB}$$

The Variational Lower Bound could be written as a sum of $\mathbb{KL}$ terms [26]:

$$L_{VLB} = \mathbb{E}_r\left[\log\frac{r(\mathbf{z}_{1:T}|\mathbf{z}_0)}{p_\psi(\mathbf{z}_{0:T}|\mathbf{z_y})}\right]$$

$$= \mathbb{E}_r\left[\log\frac{\prod_{t=1}^{T}r(\mathbf{z}_t|\mathbf{z}_{t-1})}{p(\mathbf{z}_T|\mathbf{z_y})\prod_{t=1}^{T}p_\psi(\mathbf{z}_{t-1}|\mathbf{z}_t, \mathbf{z_y})}\right] \quad \text{using Eq. (17) and (20)}$$

$$= \mathbb{E}_r\left[-\log p_\psi(\mathbf{z}_T|\mathbf{z_y}) + \sum_{t=1}^{T}\log\frac{r(\mathbf{z}_t|\mathbf{z}_{t-1})}{p_\psi(\mathbf{z}_{t-1}|\mathbf{z}_t, \mathbf{z_y})}\right]$$

$$= \mathbb{E}_r\left[-\log p_\psi(\mathbf{z}_T|\mathbf{z_y}) + \sum_{t=2}^{T}\log\frac{r(\mathbf{z}_t|\mathbf{z}_{t-1})}{p_\psi(\mathbf{z}_{t-1}|\mathbf{z}_t, \mathbf{z_y})} + \log\frac{r(\mathbf{z}_1|\mathbf{z}_0)}{r_\theta(\mathbf{z}_0|\mathbf{z}_1, \mathbf{z_y})}\right]$$

$$= \mathbb{E}_r\left[-\log p_\psi(\mathbf{z}_T|\mathbf{z_y}) + \sum_{t=2}^{T}\log\left(\frac{r(\mathbf{z}_{t-1}|\mathbf{z}_t, \mathbf{z}_0)}{p_\psi(\mathbf{z}_{t-1}|\mathbf{z}_t, \mathbf{z_y})}\cdot\frac{r(\mathbf{z}_t|\mathbf{z}_0)}{r(\mathbf{z}_{t-1}|\mathbf{z}_0)}\right) + \log\frac{r(\mathbf{z}_1|\mathbf{z}_0)}{p_\psi(\mathbf{z}_0|\mathbf{z}_1, \mathbf{z_y})}\right]$$

$$= \mathbb{E}_r\left[-\log p_\psi(\mathbf{z}_T|\mathbf{z_y}) + \sum_{t=2}^{T}\log\frac{r(\mathbf{z}_{t-1}|\mathbf{z}_t, \mathbf{z}_0)}{p_\psi(\mathbf{z}_{t-1}|\mathbf{z}_t, \mathbf{z_y})} + \sum_{t=2}^{T}\frac{r(\mathbf{z}_t|\mathbf{z}_0)}{r(\mathbf{z}_{t-1}|\mathbf{z}_0)} + \log\frac{r(\mathbf{z}_1|\mathbf{z}_0)}{p_\psi(\mathbf{z}_0|\mathbf{z}_1, \mathbf{z_y})}\right]$$

$$= \mathbb{E}_r\left[-\log p_\psi(\mathbf{z}_T|\mathbf{z_y}) + \sum_{t=2}^{T}\log\frac{r(\mathbf{z}_{t-1}|\mathbf{z}_t, \mathbf{z}_0)}{p_\psi(\mathbf{z}_{t-1}|\mathbf{z}_t, \mathbf{z_y})} + \frac{r(\mathbf{z}_T|\mathbf{z}_0)}{r(\mathbf{z}_1|\mathbf{z}_0)} + \log\frac{r(\mathbf{z}_1|\mathbf{z}_0)}{p_\psi(\mathbf{z}_0|\mathbf{z}_1, \mathbf{z_y})}\right]$$

$$= \mathbb{E}_r \left[ \log \frac{r(\mathbf{z}_T | \mathbf{z}_0)}{p_\psi(\mathbf{z}_T | \mathbf{z_y})} + \sum_{t=2}^{T} \log \frac{r(\mathbf{z}_{t-1} | \mathbf{z}_t, \mathbf{z}_0)}{p_\psi(\mathbf{z}_{t-1} | \mathbf{z}_t, \mathbf{z_y})} - \log p_\psi(\mathbf{z}_0 | \mathbf{z}_1, \mathbf{z_y}) \right]$$

$$= \mathbb{E}_r \left[ \mathbb{KL} \big[ r(\mathbf{z}_T | \mathbf{z}_0) || p_\psi(\mathbf{z}_T | \mathbf{z_y}) \big] + \sum_{t=2}^{T} KL \big[ r(\mathbf{z}_{t-1} | \mathbf{z}_t, \mathbf{z}_0) || p_\psi(\mathbf{z}_{t-1} | \mathbf{z}_t, \mathbf{z_y}) \big] - \tag{21}$$

$$\log p_\psi(\mathbf{z}_0 | \mathbf{z}_1, \mathbf{z_y}) \Big]$$

$$= \sum_{t=0}^{T} L_t \quad \text{with} \quad \begin{cases} L_0 &= -\mathbb{E}_r \Big[ \log p_\psi(\mathbf{z}_0 | \mathbf{z}_1, \mathbf{z_y}) \Big] \\ L_t &= \mathbb{E}_r \Big[ \mathbb{KL} \big[ r(\mathbf{z}_{t-1} | \mathbf{z}_t, \mathbf{z}_0) || p_\psi(\mathbf{z}_{t-1} | \mathbf{z}_t, \mathbf{z_y}) \big] \Big] \\ L_T &= \mathbb{E}_r \Big[ \mathbb{KL} \big[ r(\mathbf{z}_T | \mathbf{z}_0) || z_\psi(\mathbf{z}_T | \mathbf{z_y}) \big] \Big] \end{cases} \tag{22}$$

In the previous equations, $\mathbb{E}_r$ is a shortcut notation for $\mathbb{E}_{\mathbf{z}_{0:T} \sim r(\mathbf{z}_{0:T})}$. Note that in the optimization process, $L_T$ could be ignored because it doesn't depend on the model parameter $\psi$, this is a pure non-informative Gaussian distribution (see Eq. 22). $L_0$ is modeled by Ho et al. [47] using a separate neural network. $L_t$ is a $\mathbb{KL}$ between 2 Gaussians distributions, so it could be calculated with a closed form:

$$r(\mathbf{z}_{t-1} | \mathbf{z}_t, \mathbf{z}_0) = \mathcal{N}(\mathbf{z}_{t-1}; \tilde{\mu}_t(\mathbf{z}_t, \mathbf{z}_0), \tilde{\beta}_t \mathbf{I}) \text{ with } \begin{cases} \tilde{\mu}_t(\mathbf{z}_t, \mathbf{z}_0) &= \frac{\sqrt{\bar{\alpha}_{t-1}}\beta_t}{1 - \bar{\alpha}_t} \mathbf{z}_0 + \frac{\sqrt{\bar{\alpha}_t}(1 - \bar{\alpha}_{t-1})}{1 - \bar{\alpha}_t} \mathbf{z}_t \\ \tilde{\beta}_t &= \frac{1 - \bar{\alpha}_{t-1}}{1 - \bar{\alpha}_t} \beta_t \end{cases} \tag{23}$$

With $\tilde{\mu}_t(\mathbf{z}_t, \mathbf{z}_0)$ and $\tilde{\beta}_t \mathbf{I}$ the mean and the variance of $r(\mathbf{z}_{t-1} | \mathbf{z}_t, \mathbf{z}_0)$, respectively. Using Eq. 18 we can express $\mathbf{z}_0$ in a convenient way:

$$\mathbf{z}_0 = \frac{1}{\sqrt{\bar{\alpha}}} (\mathbf{z}_t - \sqrt{1 - \bar{\alpha}_t} \epsilon) \tag{24}$$

Therefore on can simplify $\tilde{\mu}_t(\mathbf{z}_t, \mathbf{z}_0)$ in Eq. 23:

$$\tilde{\mu}_t(\mathbf{z}_t, \mathbf{z}_0) = \tilde{\mu}_t = \frac{1}{\sqrt{\alpha_t}} \left( \mathbf{z}_t - \frac{1 - \alpha_t}{\sqrt{1 - \bar{\alpha}_t}} \epsilon \right) \tag{25}$$

Similarly, we can re-parameterize $p_\psi(\mathbf{z}_{t-1} | \mathbf{z}_t, \mathbf{z_y})$ because $\mathbf{z}_t$ is available as input at training time:

$$\mu_\psi(\mathbf{z}_t, t) = \frac{1}{\sqrt{\alpha_t}} \left( \mathbf{z}_t - \frac{1 - \alpha_t}{\sqrt{1 - \bar{\alpha}_t}} \epsilon_\psi(\mathbf{z}_t, t) \right) \tag{26}$$

One can apply the closed form formula of the $\mathbb{KL}$ between 2 gaussians distributions to compute $L_t$ in Eq. 22:

$$L_t = \mathbb{E}_r \left[ \frac{1}{2 \| \sigma_t^2 \|_2^2} \| \tilde{\mu}_t(\mathbf{z}_t, \mathbf{z}_0) - \mu_\psi(\mathbf{z}_t, t) \|_2^2 \right]$$

$$= \mathbb{E}_r \left[ \frac{1}{2 \| \sigma_t^2 \|_2^2} \left\| \frac{1}{\sqrt{\alpha_t}} \left( \mathbf{z}_t - \frac{1 - \alpha_t}{\sqrt{1 - \bar{\alpha}_t}} \epsilon \right) - \frac{1}{\sqrt{\alpha_t}} \left( \mathbf{z}_t - \frac{1 - \alpha_t}{\sqrt{1 - \bar{\alpha}_t}} \epsilon_\psi(\mathbf{z}_t, t) \right) \right\|_2^2 \right] \text{ with Eqs. 25 and 26}$$

$$= \mathbb{E}_r \left[ \frac{(1 - \alpha_t)^2}{2\alpha_t(1 - \bar{\alpha}_t) \| \sigma_t^2 \|_2^2} \| \epsilon - \epsilon_\psi(\sqrt{\bar{\alpha}_t} \mathbf{z}_0 + \sqrt{1 - \bar{\alpha}_t} \epsilon, t) \|_2^2 \right] \tag{27}$$

With further simplification of Eq. 27 [47]:

$$L_t = \mathbb{E}_r \left[ \| \epsilon - \epsilon_\psi(\sqrt{\bar{\alpha}_t} \mathbf{z}_0 + \sqrt{1 - \bar{\alpha}_t} \epsilon, t) \|_2^2 \right] \tag{28}$$

$$= \mathbb{E}_r \left[ \| \epsilon - \epsilon_\psi(\mathbf{z}_t, t) \|_2^2 \right] \tag{29}$$

### A.4.3 Architecture and Training

The DDPM model we leverage is a 1D-UNet to perform the diffusion process over the latent embeddings. The architecture of the UNet is described in Table 2:

| Network | Layer | Input Shape | Output Shape | Param # |
|---|---|---|---|---|
| **Blocks** | | | | |
| Block_MLP | Linear | $d_{in}$ | $d_{out}$ | $d_{in} * d_{out} + d_{out}$ |
| | GroupNorm | $d_{out}$ | $d_{out}$ | $2 * d_{out}$ |
| | SiLU | $d_{out}$ | $d_{out}$ | – |
| Residual | RMSNorm_MLP | $d_{in}$ | $d_{in}$ | $d_{in}$ |
| | MyAttention | $d_{in}$ | $d_{in}$ | $d_{in} * 512 + 2 * d_{in}$ |
| ResnetBlock | SiLU | $d_t$ | $d_t$ | – |
| | Linear | $d_t$ | $2 * d_{out}$ | $2 * d_{out}(d_t + 1)$ |
| | Block_MLP | $d_{in}$ | $d_{out}$ | $d_{out}(d_{in} + 3)$ |
| | Block_MLP | $d_{out}$ | $d_{out}$ | $d_{out}(d_{out} + 3)$ |
| | Identity | $d_{out}$ | $d_{out}$ | – |
| ModuleList2 | ResnetBlock | $(d_{in}, d_{in}, d_t)$ | $d_{in}$ | $2 * d_{in}(d_t + d_{in} + 4)$ |
| | ResnetBlock | $(d_{in}, d_{in}, d_t)$ | $d_{in}$ | $2 * d_{in}(d_t + d_{in} + 4)$ |
| | Residual | $d_{in}$ | $d_{in}$ | $515 * d_{in}$ |
| | Linear | $d_{in}$ | $d_{out}$ | $d_{in} * d_{out} + d_{out}$ |
| **Unet** | | | | |
| Time Embedding | SinusoidalPosEmb | [128] | [128] | – |
| | Linear | [128] | [128] | 16,512 |
| | GELU | [128] | [128] | – |
| | Linear | [128] | [128] | 16,512 |
| Downscale | Linear | [512] | [2048] | 1,050,624 |
| | ModuleList2 | [2048,128] | [1024] | 21,011,456 |
| | ModuleList2 | [1024,128] | [512] | 5,787,136 |
| | ModuleList2 | [512,128] | [256] | 1,713,920 |
| Bottleneck | ResnetBlock | [256,128] | [256] | 198,656 |
| | Residual | [256] | [256] | 131840 |
| | ResnetBlock | [256, 128] | [256] | 198656 |
| Upscale | ModuleList2 | [256,128] | [512] | 1,316,608 |
| | ModuleList2 | [512,128] | [1024] | 4,730,368 |
| | ModuleList2 | [1024,128] | [2048] | 17,849,344 |
| | ResnetBlock+Linear | [2048,128] | [2048] | 21,514,240 |
| | Linear | [2048] | [256] | 524,544 |

Table 2: The neural architecture of the diffusion model used for all experiments unless stated otherwise (the parameter count is shown for the latent size of Quickdraw-FS experiments, ie $d = 128$).

The architectures of the diffusion models for both the Quickdraw-FS and Omniglot datasets are kept identical. The only difference is that the diffusion model is applied on a latent space of size $d = 128$ for QuickDraw and of size $d = 64$ for Omniglot. The models are trained on a batch size of 128 using the DDPM scheduler for 1000 time steps. $\beta_T$ linearly spanning between $1.5 \times 10^{-3}$ and $1.95 \times 10^{-2}$ and trained for 1000 epochs. The model is optimized using the AdamW optimizer [70] with an initial learning rate of $10^{-4}$. Then we use a scheduler in which the learning rate is divided by 10 every 200 epochs.

## A.5  Impact of the regularization on the QuickDraw-FS dataset

Herein we systematically vary the $\beta$ parameter in Eq. 1 for each type of regularization and we evaluate its effect using the originality vs. recognizability framework. To visualize this effect while maintaining the order of the hyper-parameters, we use the parametric fit method described in [54]. This technic involves 2 simultaneous parametric fit: i) a polynomial fit (degree 2) between the hyperparameters and the originality values (shown in Fig. A.4b, Fig. A.5b, Fig. A.6b, Fig. A.8b and Fig. A.9b) and ii) another a polynomial fit (degree 2) between the hyperparameters and the recognizability values (shown in Fig. A.4c, Fig. A.5c, Fig. A.6c, Fig. A.8c and Fig. A.9c). Those 2 fits could then be combined to create an oriented parametric fit between the originality and the recognizability (shown in Fig. A.4a, Fig. A.5a, Fig. A.6a, Fig. A.8a and Fig. A.9a). In these curves, the "chevron" indicates the direction in which the value of the $\beta$ hyperparameter is increased. We have included the range of $\beta$ we have explored in the caption of each type of regularized LDM. We use the notation $[a : b :: c]$ to express that we explored from $a$ to $b$ with a step of $c$.

### A.5.1  Impact of the KL regularization

Herein we evaluate a LDM leveraging a RAE trained with the following loss (with $\mathcal{L}_{KL}$) in Eq. 2:

$$\min_{\theta,\phi} \mathcal{L}_{RAE} \quad \text{s.t.} \quad \mathcal{L}_{RAE} = -\mathbb{E}_{\mathbf{z}\sim q_\phi(.|\mathbf{x})}\left[\log p_\theta(\mathbf{x}|\mathbf{z})\right] + \beta_{KL}\mathcal{L}_{KL}(\mathbf{z}) \tag{30}$$

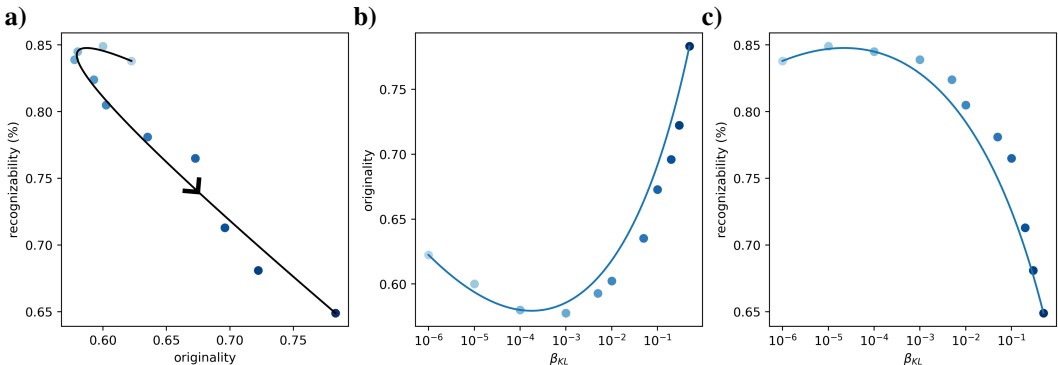

Figure A.4: **Impact of the $\beta_{KL}$ hyperparameter on the originality vs. recognizability.** Each data point corresponds to a LDM trained with a different value of $\beta_{KL}$ in Eq. 30. Herein we have explored the following $\beta_{KL}$ range : $[10^{-6}\!:\!10^{-2}\!::\!10^{-1}]$ and 0.05 and $[0.1\!:\!0.5\!::\!0.1]$.

### A.5.2  Impact of the VQ regularization

Herein we evaluate a LDM leveraging a RAE trained with the following loss (with $\mathcal{L}_{VQ}$) in Eq. 3:

$$\min_{\theta,\phi} \mathcal{L}_{RAE} \quad \text{s.t.} \quad \mathcal{L}_{RAE} = -\mathbb{E}_{\mathbf{z}\sim q_\phi(.|\mathbf{x})}\left[\log p_\theta(\mathbf{x}|\mathbf{z})\right] + \beta_{VQ}\mathcal{L}_{VQ}(\mathbf{z}) \tag{31}$$

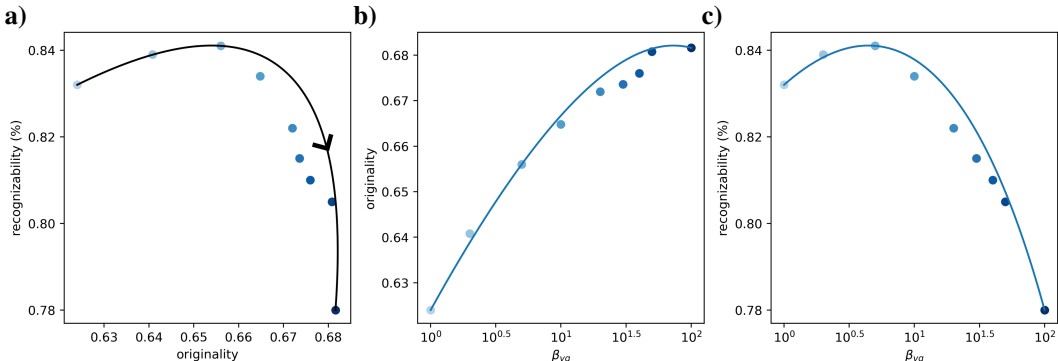

Figure A.5: **Impact of the $\beta_{VQ}$ hyperparameter on the originality vs. recognizability.** Each data point corresponds to a LDM trained with a different value of $\beta_{VQ}$ in Eq. 31. Herein we have explored the following $\beta_{VQ}$ range : $[1, 2, 5]$ and $[10\!:\!50\!::\!10]$ and 100.

### A.5.3 Impact of the CL regularization

Herein we evaluate a LDM leveraging a RAE trained with the following loss (with $\mathcal{L}_{CL}$) in Eq. 4:

$$\min_{\theta,\phi} \mathcal{L}_{RAE} \quad \text{s.t.} \quad \mathcal{L}_{RAE} = -\mathbb{E}_{\mathbf{z} \sim q_\phi(.|\mathbf{x})} \left[\log p_\theta(\mathbf{x}|\mathbf{z})\right] + \beta_{CL}\mathcal{L}_{CL}(\mathbf{z}) \tag{32}$$

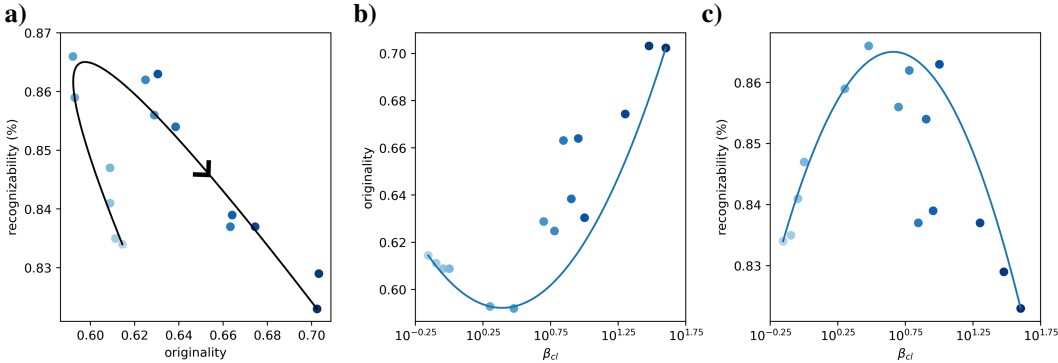

Figure A.6: **Impact of the $\beta_{CL}$ hyperparameter on the originality vs. recognizability.** Each data point corresponds to a LDM trained with a different value of $\beta_{CL}$ in Eq. 32. Herein we have explored the following $\beta_{CL}$ range : $[0.7\!:\!0.9\!::\!0.1]$ and $[1\!:\!10\!::\!1]$ and $[10\!:\!40\!::\!10]$.

### A.5.4 Impact of the prototype-based regularization

Herein we evaluate a LDM leveraging a RAE trained with the following loss (with $\mathcal{L}_{PR}$) in Eq. 5:

$$\min_{\theta,\phi} \mathcal{L}_{RAE} \quad \text{s.t.} \quad \mathcal{L}_{RAE} = -\mathbb{E}_{\mathbf{z} \sim q_\phi(.|\mathbf{x})} \left[\log p_\theta(\mathbf{x}|\mathbf{z})\right] + \beta_{PR}\mathcal{L}_{PR}(\mathbf{z}) \tag{33}$$

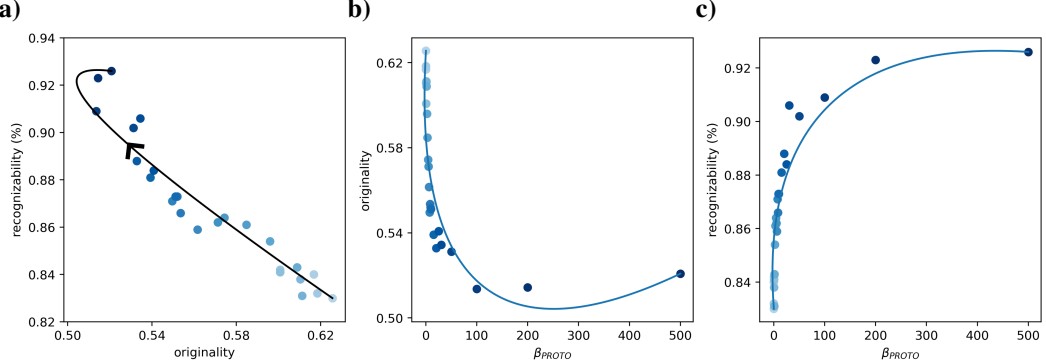

Figure A.7: **Impact of the $\beta_{PR}$ hyperparameter on the originality vs. recognizability.** Each data point corresponds to a LDM trained with a different value of $\beta_{PR}$ in Eq. 33. Herein we have explored the following $\beta_{PR}$ range : $[10^{-4}\!:\!10^{-1}\!::\!10^{-1}]$ and $[0.25\!:\!0.75\!::\!0.25]$ and $[1.0\!:\!10\!::\!1]$ and $[15\!:\!30\!::\!5]$ and $[100, 200, 500]$.

### A.5.5   Impact of the SimCLR regularization

Herein we evaluate a LDM leveraging a RAE trained with the following loss (with $\mathcal{L}_{SimCLR}$) in Eq. 14:

$$\min_{\theta,\phi} \mathcal{L}_{RAE} \quad \text{s.t.} \quad \mathcal{L}_{RAE} = -\mathbb{E}_{\mathbf{z}\sim q_\phi(.|\mathbf{x})}\left[\log p_\theta(\mathbf{x}|\mathbf{z})\right] + \beta_{SimCLR}\mathcal{L}_{SimCLR}(\mathbf{z}) \tag{34}$$

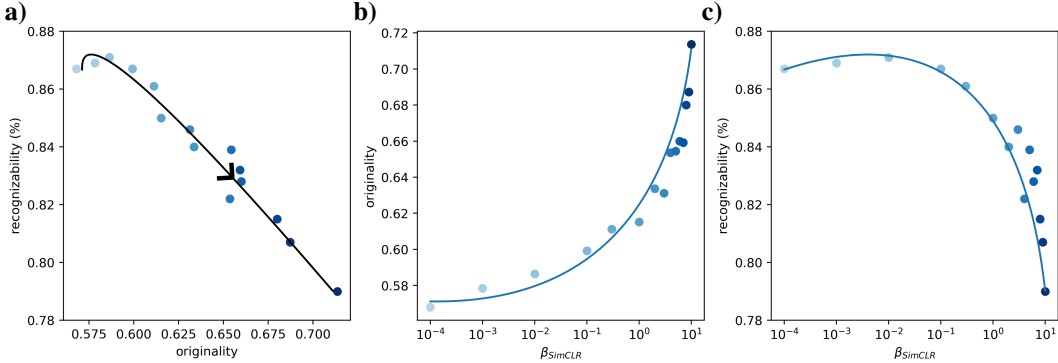

Figure A.8: **Impact of the $\beta_{SimCLR}$ hyperparameter on the originality vs. recognizability.** Each data point corresponds to a LDM trained with a different value of $\beta_{SimCLR}$ in Eq. 34. Herein we have explored the following $\beta_{SimCLR}$ range : $[10^{-4}\!:\!10^{-1}\!::\!10^{-1}]$ and $[1\!:\!10\!::\!1]$.

### A.5.6   Impact of the Barlow regularization

Herein we evaluate a LDM leveraging a RAE trained with the following loss (with $\mathcal{L}_{BAR}$) in Eq. 15:

$$\min_{\theta,\phi} \mathcal{L}_{RAE} \quad \text{s.t.} \quad \mathcal{L}_{RAE} = -\mathbb{E}_{\mathbf{z}\sim q_\phi(.|\mathbf{x})}\left[\log p_\theta(\mathbf{x}|\mathbf{z})\right] + \beta_{BAR}\mathcal{L}_{BAR}(\mathbf{z}) \tag{35}$$

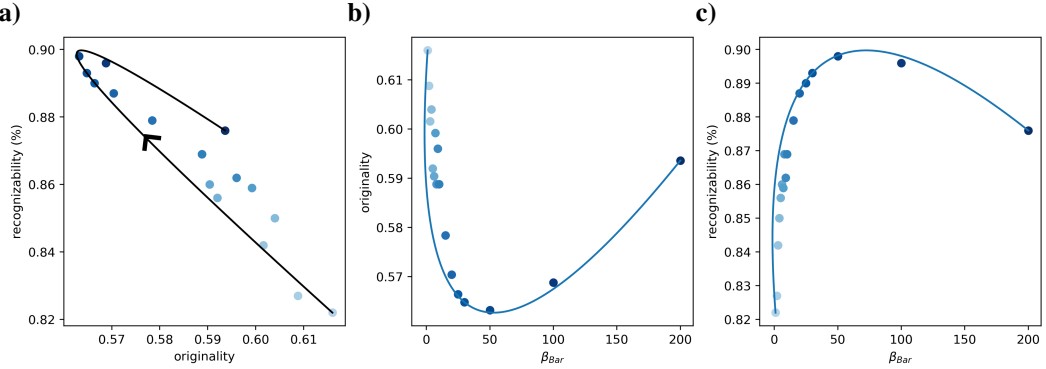

Figure A.9: **Impact of the $\beta_{BAR}$ hyperparameter on the originality vs. recognizability.** Each data point corresponds to a LDM trained with a different value of $\beta_{BAR}$ in Eq. 35. Herein we have explored the following $\beta_{BAR}$ range : $[1\!:\!10\!:\!:\!1]$ and $[15\!:\!30\!:\!:\!5]$ and $[50, 100, 200]$.

## A.6 Impact of the regularization on the Omniglot dataset dataset

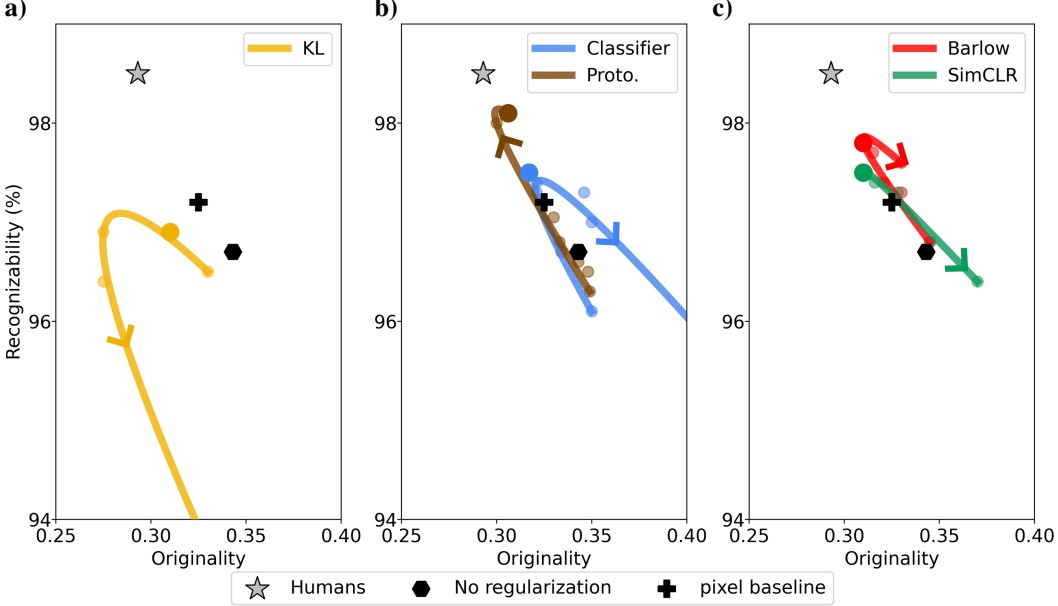

Figure A.10: **Effect of increasing the regularization weights on the originality vs recognizability framework (Omniglot dataset).** Each data point represents an LDM trained with different values of regularization weights ($\beta$). The curves represent the parametric fits, oriented in the direction of an increase of $\beta$. **a):** For the LDMs with "standard" regularizers, the $\beta$ is applied on the **KL** ($\mathcal{L}_{KL}$ in Eq. 2). **b):** For the supervised regularizations, the $\beta$ is applied on the **CL** ($\mathcal{L}_{CL}$ in Eq. 4) or on the **prototype**-based regularizations ($\mathcal{L}_{PR}$ in Eq. 5). **c):** For the contrastive regularizations, the $\beta$ is applied on the **SimCLR** ($\mathcal{L}_{SimCLR}$ in Eq. 14) or on the **Barlow** regularizations ($\mathcal{L}_{Bar}$ in Eq. 15). See A.5 for more information on the range of $\beta$ we have explored for each regularization. Larger data points indicate models whose performance is closer to that of humans for each type of regularization. For comparison, we include a LDM leveraging a non-regularized RAE (hexagon marker) and a diffusion model trained directly on the pixel space (cross marker). The human performance corresponds to the recognizability and originality of human drawings (shown with a grey star)

Here we present a curve similar to Fig. 3 but for LDMs trained on the Omniglot dataset. We were unable to train a VQ-VAE with reasonable performance on this dataset, so we have excluded the **VQ**-regularized LDM from Fig. A.10. We believe this issue is due to improper hyperparameter tuning as the same regularizer works reasonably well on the QuickDraw-FQ dataset. We are actively working to resolve this problem.

Except for the **VQ** regularizer, we observe that all other regularizers follow a similar trend to those trained on the QuickDraw-FS dataset. In particular, the **prototype**-based and the **Barlow** regularizers outperform all others.

### A.7   Samples generated by the one-shot LDMs

Here we showcase the images generated by one-shot LDMs. The exemplars used to condition the LDMs are present in top line in the red frame. We randomly chose 10 exemplars from 115 possible options in the QuickDraw-FS test set. All images below the red frame represent samples of the corresponding visual concept generated by the LDM. We use the same 10 exemplars for all the LDMs for easy comparison. All shown exemplar corresponds to the LDMs, for each regularizer, showing the shortest distance to humans. They correspond to larger data points in Fig. 3.

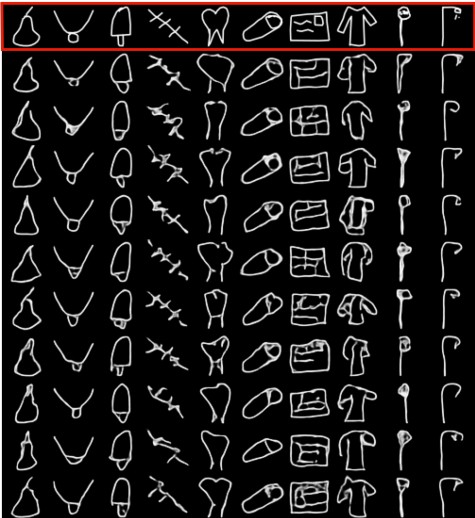

Figure A.11: **Samples generated by a LDM without regularzation**. For this LDM, $\beta$ is set to 0.

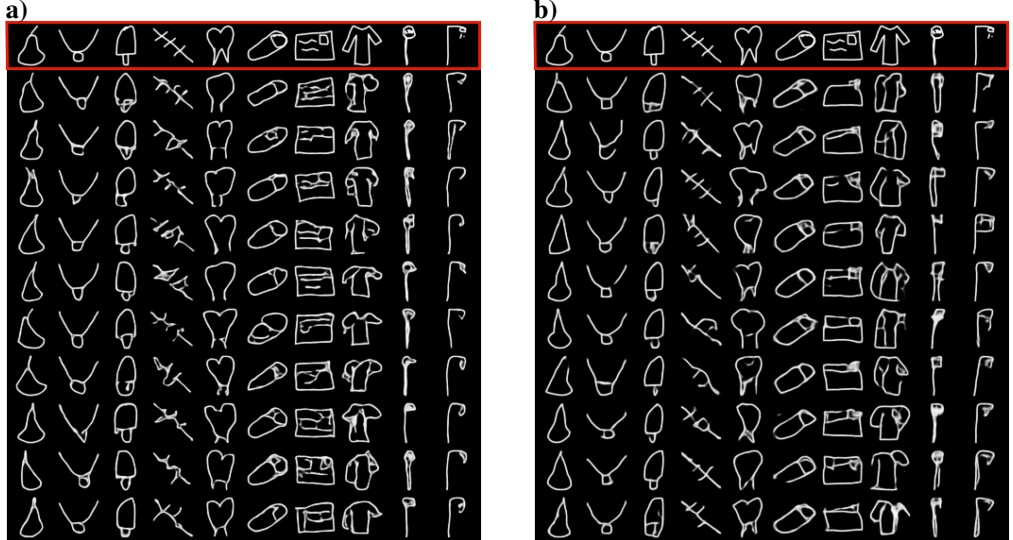

Figure A.12: **Samples generated by LDMs with standard regularizer. a)** **KL** regularizer (obtained with $\beta_{KL} = 10^{-5}$). **b)** **VQ** regularizer (obtained with $\beta_{VQ} = 5$).

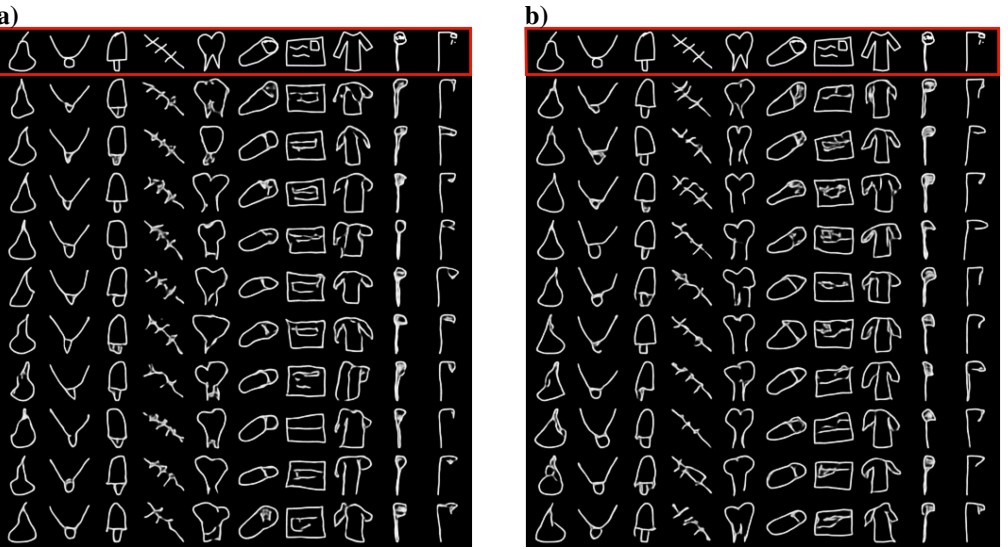

Figure A.13: **Samples generated by LDMs with supervised regularizers. a)** **classification** regularizer (obtained with $\beta_{CL} = 5$). **b)** **prototype**-based regularizer (obtained with $\beta_{PR} = 5 \cdot 10^{2}$).

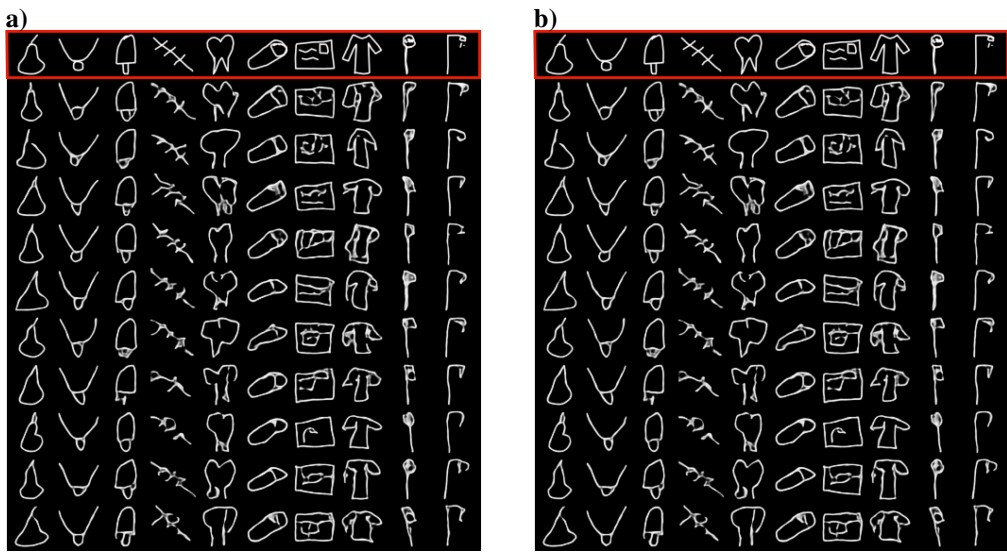

Figure A.14: **Samples generated by LDMs with contrastive regularizer. a) SimCLR** regularizer (obtained with $\beta_{SimCLR} = 10^{-2}$). **b) Barlow** regularizer (obtained with $\beta_{BAR} = 30$).

### A.8 LDM feature importance maps

#### A.8.1 Mathematics behind the feature importance maps

We remind that $p_\theta(\mathbf{x}|\mathbf{z})$ is the decoder of the RAE, and that $p_\psi(\mathbf{z_{t-1}}|\mathbf{z_t}, \mathbf{z_y})$ is the transition probability learned by the diffustion model. To make the mathematical derivations more concise, we define the following function :

$$p_\theta : \mathbb{R}^d \longrightarrow \mathbb{R}^D \qquad \text{and} \qquad p_\psi : \mathbb{R}^d \longrightarrow \mathbb{R}^d \qquad (36)$$

$$\mathbf{z} \longmapsto \mathbf{x} = \log p_\theta(\cdot|\mathbf{z}) \qquad\qquad \mathbf{z_t} \longmapsto \mathbf{z_{t-1}} = \log p_\psi(\cdot|\mathbf{z_t}, \mathbf{z_y}) \quad (37)$$

To project each intermediate noisy state $\mathbf{z_t}$ into the pixel, we feed them into the decoder. The resulting projection is $\mathbf{x_t} = p_{\theta,\psi}(\mathbf{z_t}) = p_\theta \circ p_\psi(\mathbf{z_t})$

For each time step of the diffusion process, the importance feature map quantifies how the absolute value of $p_{\theta,\psi}(\mathbf{z_t})$ changes when one varies $\mathbf{z_t}$. $\phi(\mathbf{x}, \mathbf{y})$ describes the accumulation, over all time steps, of these "local feature map":

$$\phi(\mathbf{x}, \mathbf{y}) = \sum_{t=0}^{T} \left| \frac{\partial p_{\theta,\psi}(\mathbf{z_t})}{\partial \mathbf{z_t}} \right| \qquad (38)$$

$$= \sum_{t=0}^{T} \left| \frac{\partial p_\theta \circ p_\psi(\mathbf{z_t})}{\partial \mathbf{z_t}} \right| \qquad (39)$$

$$= \sum_{t=0}^{T} \left| \frac{\partial p_\theta}{\partial \mathbf{x_t}}(p_\psi(\mathbf{z_t})) \frac{\partial p_\psi}{\partial \mathbf{z_t}}(\mathbf{z_t}) \right| \qquad (40)$$

$$= \sum_{t=0}^{T} \left| J_{p_\theta}(\mathbf{x_t}) \nabla_{\mathbf{z_t}} p_\psi(\mathbf{z_t}) \right| \qquad (41)$$

$$(42)$$

with $J_{p_\theta}(\mathbf{x_t})$ the Jacobian of the function $p_\theta$ w.r.t $\mathbf{x_t}$ computed in $p_\psi(\mathbf{z_t})$. If we trade the functional notations for probabilistic ones we have:

$$\phi(\mathbf{x}, \mathbf{y}) = \sum_{t=0}^{T} \left| J_{\log p_\theta(\cdot|\mathbf{z_t})}(\mathbf{x_t}) \nabla_{\mathbf{z_t}} \log p_\psi(\cdot|\mathbf{z_t}, \mathbf{z_y}) \right| \qquad (43)$$

### A.8.2 Example of LDM feature importance maps

The LDMs' feature importance maps have been computed on 25 different categories, for each of the six different regularization methods discussed in the paper. The feature maps were calculated by taking the average of $n = 10$ misalignment maps $\phi(\mathbf{x}, \mathbf{y})$ as defined in Eq. 9. All shown feature importance maps correspond to the LDMs, for each regularizer, showing the shortest distance to humans. They correspond to larger data points in Fig. 3.

a)            b)

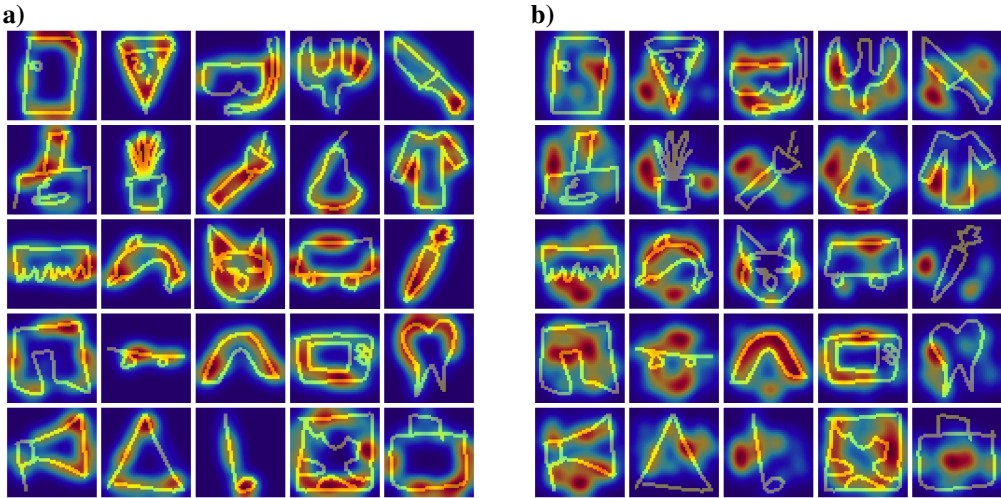

Figure A.15: **Feature importance maps for LDMs with standard regularizer. a) KL** regularizer (obtained with $\beta_{KL} = 10^{-5}$). **b) VQ** regularizer (obtained with $\beta_{VQ} = 5$).

a)            b)

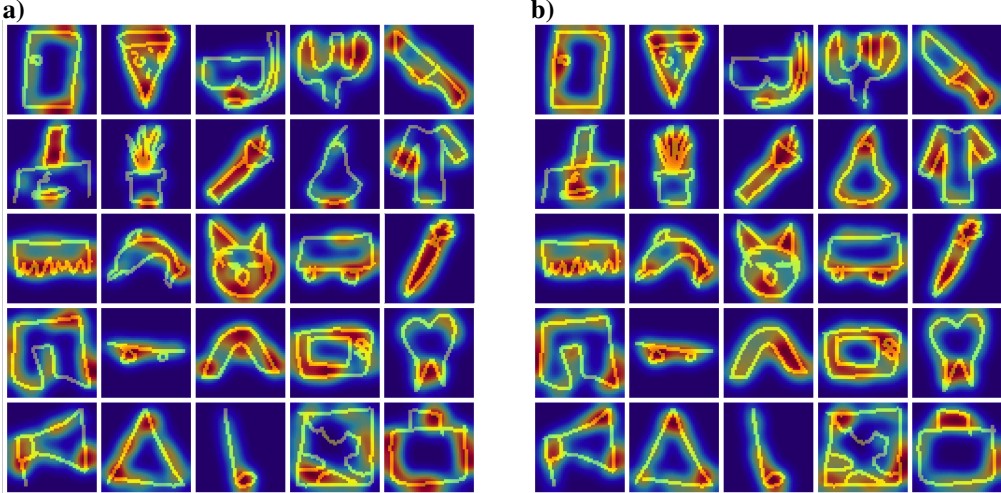

Figure A.16: **Feature importance maps for LDMs with supervised regularizer. a) classification** regularizer (obtained with $\beta_{CL} = 5$). **b) prototype**-based regularizer (obtained with $\beta_{PR} = 5 \cdot 10^2$).

a) 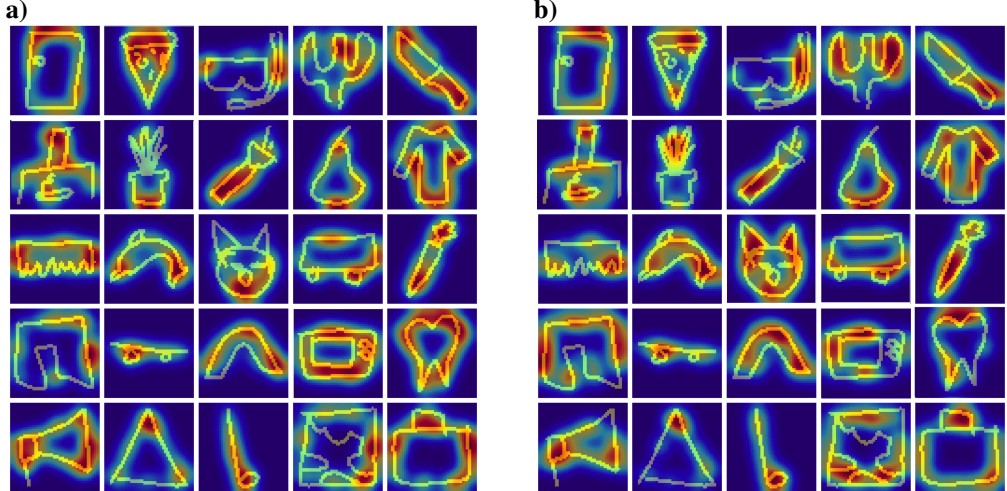 b)

Figure A.17: **Feature importance maps for LDMs with contrastive regularizer. a) SimCLR** regularizer (obtained with $\beta_{SimCLR} = 10^{-2}$). **b) Barlow** regularizer (obtained with $\beta_{BAR} = 30$).

### A.8.3 Example of Human feature importance maps

For comparison, feature importance maps have also been computed for humans for the same 25 categories. For humans, the feature importance maps are heatmaps representing the likelihood of a pixel being selected by a participant as part of the ClickMe-QuickDraw experiment (further details on the experiment provided in App. S of Boutin et al. [30]). The same image used to calculate the misalignment maps for the LDMs is presented to the participants during the CliCkMe-QuickDraw experiment.

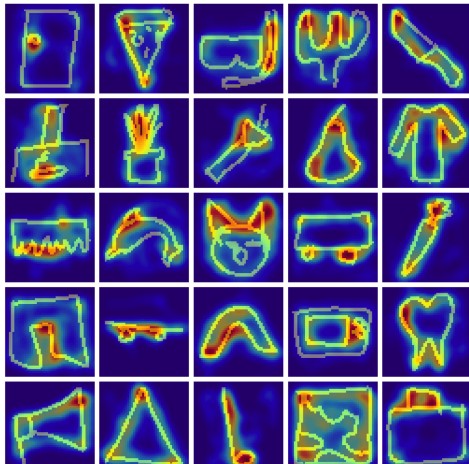

Figure A.18: **Feature Importance maps for humans**

**Human consistency:** To evaluate how humans agree with each other on the feature importance maps, we computed the human consistency. To do so we use a bootstrapping technique. For each category, we divided the participants into 2 populations (randomly selected), obtaining approximately 25 annotations (heatmaps) coming from different participants for each category. We then average those annotations within the same population (and the same category) to form population-wise feature importance maps. We finally compute the human consistency with the Spearman correlation between those population-wise feature importance maps. We obtain a spearman of $0.8845$ ($p < 5.10^{-2}$).

### A.8.4 Pair-wise statistical test for importance feature maps

To verify the statistical significance between the human/machine correlation we have obtained for all types of regularized LDMs we use a pair-wise statistical test. In particular, we compute the Wilcoxon signed-rank test between all pairs of LDMs. This test is non-parametric and does not consider the "Gaussianity" of the underlying population. The null hypothesis of this test (that could not be rejected when the $p$-value is over $0.05$) is that the two tested populations are sampled from the same distribution. The alternative hypothesis (validated when the $p$-value is below $0.05$) is that the first population ( columns of the Table A.8.4) is stochastically greater than the second population (rows of the Table A.8.4). All $p$-values, for all pairwise statistical tests are shown in Table A.8.4.

|  | Barlow | SimCLR | Classif. | KL | VQ | No reg. |
|---|---|---|---|---|---|---|
| **Proto.** | $5.4 \times 10^{-4}$ | $5.9 \times 10^{-6}$ | $6.03 \times 10^{-5}$ | $1.2 \times 10^{-6}$ | $2.3 \times 10^{-7}$ | $4.7 \times 10^{-7}$ |
| **Barlow** |  | $9.5 \times 10^{-4}$ | $9.5 \times 10^{-4}$ | $1.8 \times 10^{-4}$ | $2.3 \times 10^{-7}$ | $2.3 \times 10^{-7}$ |
| **SimCLR** |  |  | $2.3 \times 10^{-1}$ | $5.2 \times 10^{-2}$ | $4.7 \times 10^{-7}$ | $4.7 \times 10^{-7}$ |
| **Classif** |  |  |  | $2.9 \times 10^{-1}$ | $2.3 \times 10^{-7}$ | $2.3 \times 10^{-7}$ |
| **KL** |  |  |  |  | $2.3 \times 10^{-7}$ | $4.5 \times 10^{-6}$ |
| **VQ** |  |  |  |  |  | $9.9 \times 10^{-1}$ |

Importantly those statistical tests have been computed on the Spearman correlation vector (one Spearman value per category) between the feature importance maps of the best-performing models (those indicated with bigger data points in Fig. 3) and those of humans.

### A.8.5 Illustration of the limited one-shot ability of Dall-e

Herein we illustrate how current Latent Diffusion Models tend to fail at producing faithful variations when prompted with a single image. We showcase some of the generations made by Dall-e 3 when conditioned on a single image of a self-balancing bike. The self-balancing bike is a particularly interesting use case as it represents an 'unusual' vehicle that is unlikely to belong to the Dall-e 3 training database. You can observe that Dall-e generates images missing some of the key concepts of the self-balancing bike (i.e. one-wheel).

**Exemplar**          **Variations**

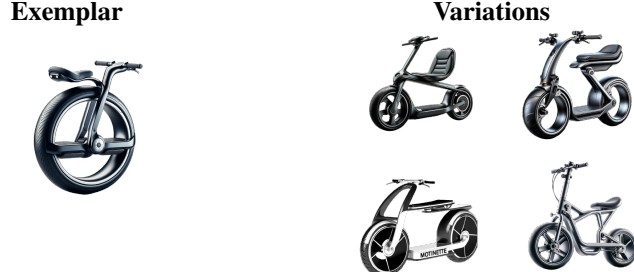

Figure A.19: **Examples of variations** generated by Dall-e 3 when prompted with a single image of a self-balancing bike

### A.8.6 Potential limitations

In this article, we tested six representational inductive biases, a small number considering the extensive range available in the representation-learning literature. This field encompasses hundreds of inductive biases that have proven successful in one-shot classification tasks. Therefore, other representational inductive biases might align better with human performance, both in terms of sample similarity and visual strategy. Our goal wasn't to test all possible biases but to demonstrate that some of them can significantly narrow the gap with humans in one-shot drawing tasks.

Another limitation of this article lies in the recognizability vs. originality framework we are using to evaluate the drawings. This framework leverages 2 critic networks to evaluate the sample's originality and recognizability. There's no guarantee these networks align with human perceptual judgments. Thus, the recognizability and originality scores might not reflect human perception accurately. However, since both human and model outputs are evaluated using the same pre-trained critic networks, the comparison remains fair.

Our approach leveraged two-stage generative models: the first stage compresses information and shapes the latent distribution with representational inductive biases (the RAE), and the second stage learns this latent distribution (the diffusion model). This type of architecture takes longer to train because it requires two separate training procedures. However, this limitation could be overcome by using an end-to-end training procedure for Latent Diffusion Models, which could streamline the process [71].

### A.8.7 Computational Resources

All the experiments of this paper have been performed using Quadro-RTX600 GPUs with 16 GB memory. The training time for the RAE is approximately 24 hours and 72 hours for the Diffusion model (96 hours overall). Note that as we have explored a large range of hyperparameters for all types of regularization, our paper is relatively extensive in terms of computations (600 models have been trained overall, but just a small part of them have been used in this article).

### A.8.8 Broader Impact

This work does not present any foreseeable negative societal consequences. We think the societal impact of this work is positive. It might help the neuroscience community to evaluate the different mechanisms that allow human-level generalization and then better understand the brain.

