# OpenReview forum: "Latent Representation Matters: Human-like Sketches in One-shot Drawing Tasks"
_NeurIPS.cc/2024/Conference — NeurIPS 2024 poster_

### Official Review · Reviewer_YDjR · 2024-07-12

**Soundness:** 3
**Presentation:** 4
**Contribution:** 2
**Rating:** 6
**Confidence:** 2

**Summary:**

This paper investigates how different regularization techniques, applied to the latent space of Latent Diffusion Models (LDMs), impact their performance on one-shot drawing tasks. The Authors explore many regularization methods: KL divergence, vector quantization, classification, prototype-based, SimCLR, and Barlow Twins.

They evaluate these methods against human performance using quantitative metrics (originality vs. recognizability) and qualitative analysis of feature importance maps.

The results show that LDMs with prototype-based and Barlow Twins regularizations produce sketches that are most similar to human drawings in terms of both recognizability and originality.

**Strengths:**

- The paper provides a comprehensive comparison thorough examination of six different regularization techniques, offering insights into their effectiveness for one-shot drawing tasks.

- It introduces a novel method for generating feature importance maps in LDMs, allowing for direct comparison with human perceptual strategies.

- From the practical side, the findings have potential applications in improving generative models for tasks requiring human-like generalization abilities.

- The study has an highly interdisciplinary approach since it integrates computer science, cognitive science, and neuroscience, potentially offering insights into human visual cognition.

The paper is very clearly written, and the Authors provide detailed information about their experimental setup, hyperparameters, and code availability, enhancing reproducibility.

**Weaknesses:**

Overall, I think that the paper makes a nice contribution to our understanding of how different regularization techniques affect the latent representations in generative models and their ability to produce human-like sketches.

The weakness I see regards the possibility to generalize from the "simple" datasets analyzed to more complex creative processes.
This study primarily focuses on the QuickDraw-FS dataset, with limited exploration of the Omniglot dataset. It not very clear to me how sound can be the extrapolation from these very simple (although relevant) contexts to more complex ones. I know that this does not provide a concrete and actionable insight, but I would appreciate a comment on this.

**Questions:**

How generalizable are these findings to more complex drawings?

**Limitations:**

Limitations are adequately discussed, but they are not part of the main text (they are discussed in the Appendix, pag. 34).

---

> ### Author Rebuttal · Authors · 2024-08-05
>
> We thank the reviewer YDjR for the positive feedback as well as the relevant comments. We especially appreciated that the reviewer highlighted the interdisciplinary approach, which was at the heart of the article. Unfortunately, interdisciplinarity has its own limitations, especially when it comes to comparing humans and machines. This is the main reason why we have limited ourselves to rather simple datasets. We detail our answer below:
>
> * **About the generalization to other and more complex datasets**: We want to clarify that our experiments on the Omniglot datasets are actually not 'limited'. We placed the Omniglot results in the supplementary information section to keep a concise main-text article, but the Omniglot analysis is consistent with that conducted on the QuickDraw Dataset. In this article, our focus has been primarily on the one-shot drawing task because it allows fair comparison between humans and machines; more complex tasks like natural image generation by contrast are beyond human capability. Nevertheless, the latent diffusion models and the regularizers we have used in the article are known to scale well on more complex datasets (e.g. [1]). We anticipate similar performance improvements on natural image datasets as observed with the QuickDraw data, especially since regularizers like Barlow, SimCLR, and prototypical have shown strong performance in one-shot classification tasks of natural images. However such natural image generation models won’t be comparable to human performance as humans can hardly synthesize such images.
>
>
> * **About the limitations**: We agree that the limitation should be in the main text and not in the appendix. We have therefore added a paragraph summarizing our limitation in the discussion section (line 351)
>
>
> Overall, we think our response has addressed the primary concern of the reviewer, clarifying that our deliberate choice of a relatively simple one-shot drawing setting allows us to draw a fair comparison between humans and machines to effectively answer our scientific question. We hope we have convinced the reviewer to increase their rating. Should there be any remaining issues, we are more than willing to engage in further discussion.
>
>
> [1] Rombach, Robin, et al. "High-resolution image synthesis with latent diffusion models." Proceedings of the IEEE/CVF conference on computer vision and pattern recognition. 2022.

---

> ### Author Response · Authors · 2024-08-12
>
> Should there be any remaining issues, we are more than willing to engage in further discussion.

---

### Official Review · Reviewer_1Pi8 · 2024-07-12

**Soundness:** 2
**Presentation:** 3
**Contribution:** 2
**Rating:** 4
**Confidence:** 5

**Summary:**

The authors propose to explore how different regularizers impact the performance of LDM on one-shot sketch generation, with a specific focus on evaluating the similarity between the generated sketches and real ones. It reveals that prototype- and Barlow-based regularizers are among the best, and claims the gap between humans and machines in the one-shot drawing task is almost closed.

**Strengths:**

- This seems the first time that the latent diffusion model (LDM) is applied to one-shot sketch generation.
- The authors discussed how different regularizers impact the generation results regarding the originality (diversity) and recognizablity, which is valuable. Interestingly, prototype- and Barlow-based approaches are the best.
- The paper is well-written and easy to follow.

**Weaknesses:**

- This paper is more like an incremental work based on [30], the key idea of using the diversity vs recognizability framework, and importance maps to measure the generation quality of diffusion models is the same. It differs in extending the diffusion model into latent feature space and applying different regularizers, which seems minor.
- It is a bit over-claimed that the gap between humans and machines is almost closed on the one-shot drawing task by using LDM plus proper regularizers. The qualitative results shown are not as good as the actual sketches, suffering from clear blur and distortion. The experiments are conducted on relatively simple sketch cases, which makes it hard to justify its scalability.
- DDPM used in [30] for this task is not compared (with or without the same regularizers if possible), it would be helpful to understand how effectiveness of pushing the denoising into latent space.

**Questions:**

- It is unclear how to construct the sketch codebook used for VQ-VAE.

**Limitations:**

Please refer to the weaknesses.

---

> ### Author Rebuttal · Authors · 2024-08-05
>
> We thank the reviewer 1Pi8 for the meaningful comments. Please find below a point-by-point response that addresses the reviewer’s concerns:
>
> * **About the incremental work compared to [30]** : We agree with the reviewer that our article builds on the comparison framework and some ideas introduced in [30], but we respectfully disagree that our article does not bring significant novelties compared to [30]. Here are the 2 main novelties:
>     * While [30] focuses on comparing various types of generative models (GANs, VAEs, and diffusion models), we focus on comparing the effect of inductive biases (through regularization) in the latent space of latent-diffusion models. This difference might seem minor to the reviewer, but this question of effective inductive biases is prevalent in cognitive psychology (and is still an open question, see [1, 2] below) and has never been systematically studied in latent-diffusion models from a machine learning perspective. Our results confirm how crucial such inductive biases are as it has a strong impact on the generalization performance of the one-shot drawing setting.
>     * The method used in [30] to generate feature importance maps leads to maps in which the background is dominant (see Fig 5 in [30]), preventing the authors from making any meaningful and quantifiable comparison with human feature importance maps. In our article, we introduce a novel method to derive importance maps. In contrast to [30] our method leads to importance maps that are directly comparable with those of humans. We think this is an important difference with [30] as our results are backed by two independent methods (the recognizability from originality methods, and the importance maps comparison), strengthening our claim.
>
>   To clarify and emphasize the differences with [30] we have added a few extra sentences to summarize those 2 major differences in the related work section (lines 129-130).
>
>
> * **About the claim that “the gap between humans and machines is almost close” that might be a bit overstated**: We agree with the reviewer that this claim may be somewhat overstated, considering that our comparison between humans and machines is based on a very specific task, which does not cover the full spectrum of abilities in both humans and machines. We have therefore downplayed this claim and we have replaced it in the narrower context of our article (in the abstract, and in the discussion section).
>
>
> * **About the comparison with the DDPM**: We have actually included the DDPM from [30] in our comparison, this is what we call the pixel baseline (see Fig 2). Note that our baseline is a guided version of the DDPM because previous articles have shown that guided DDPM performs better than their non-guided counterpart in the one-shot generation task. We call it ‘pixel-baseline’ , because the DDPM is directly applied in the pixel space in contrast to other latent diffusion models that are applied in a regularized latent space. But we agree with the reviewer that this designation is rather ambiguous. We have therefore changed it to ‘pixel-space DDPM’.
>
> * **About the sketch codebook of the VQ-VAE**: The codebook in the VQ-VAE could be viewed as a dictionary of vectors, with each vector being learned so that it minimizes the L2 distance with the latent coordinate. Note that the way we learn the codebook is similar to the standard procedure described in the original VQ-VAE article [3]. To be more concrete, let’s consider the case where the latent space (before discretization) is of size (4, 128) (here we ignore the batch dimension for the sake of concision). Let’s also consider we have a codebook with 512 elements (this is the codebook size we use in the article). In this case, all 512 vectors of the codebook will be of size 4, so that they match the number of channels of the latent space. During learning, the codebook vectors are learned so that they minimize the L2 distance between 128 vectors (of size 4) of the latent space (see section A.2.1. for a pseudo-code). During inference, the discretization process associates to each of the 128 vectors of the latent space the address (i.e. an integer) of the closest vector in the notebook. This allows the transformation of the continuous-valued latent space into a discretized one. We agree with the reviewer that a clear explanation of codebook learning is missing in the article. We have included a scheme to explain this process in section A.2.1 to clarify this point.
>
> Overall, we hope that our detailed response has addressed the reviewer's concerns, and convinced them to increase their rating. If some concerns remain, we will be pleased to engage into more discussion.
>
>
> [1] Goyal, Anirudh, and Yoshua Bengio. "Inductive biases for deep learning of higher-level cognition." Proceedings of the Royal Society A 478.2266 (2022): 20210068. \
> [2] Marjieh, Raja, et al. "Using Contrastive Learning with Generative Similarity to Learn Spaces that Capture Human Inductive Biases." arXiv preprint arXiv:2405.19420 (2024). \
> [3] Van Den Oord, Aaron, and Oriol Vinyals. "Neural discrete representation learning." Advances in neural information processing systems 30 (2017). \
> [30] Boutin, Victor, et al. "Diffusion models as artists: are we closing the gap between humans and machines?." Proceedings of the 40th International Conference on Machine Learning. 2023.

---

> ### Author Response · Authors · 2024-08-12
>
> Should there be any remaining issues, we are more than willing to engage in further discussion.

---

### Official Review · Reviewer_ZReU · 2024-07-12

**Soundness:** 2
**Presentation:** 3
**Contribution:** 2
**Rating:** 4
**Confidence:** 4

**Summary:**

i)This paper uncovers the impact of representational inductive biases on Latent Diffusion Models through one-shot tasks, particularly in the realm of human-like sketching. It compares three distinct groups of regularizers: a standard baseline, supervised methods, and a third group consisting of self-supervised techniques.

ii)This paper aims to uncover the strategies employed by each regularization method to generalize to novel categories.

iii)This paper conducts a comprehensive comparative analysis of the effectiveness of different regularization methods in one-shot drawing tasks. The study explores various dimensions, examining the performance and differences of various regularization strategies in such tasks.

**Strengths:**

This paper is dedicated to analyzing the specific manifestations of various inductive biases in one-shot drawing tasks, contributing new findings and demonstrating a high level of originality. The paper is well written, with a clear and logical structure, reflecting a rigorous academic attitude. The experimental section is well-designed, with substantial research effort, and the execution process strictly adheres to scientific methods. The comparative analysis strategy used is both comprehensive and detailed, effectively ensuring the precision and credibility of the research findings. Additionally, the conclusions of this paper provide valuable insights for the field of one-shot drawing tasks and have significant guiding significance for subsequent research.

**Weaknesses:**

i)This paper conducted an extensive analysis of the impact of different inductive biases in one-shot drawing tasks. However, the analysis remains superficial, merely briefly revealing the experimental results of various methods without delving into the fundamental reasons behind the differences in outcomes among the methods. Furthermore, the paper does not propose specific solutions to the research questions addressed.

ii)The experimental method adopted in this study is not limited to specific modalities and is not restricted by data scale. This method may be applicable to the generation tasks of more modalities(such as photos or text2img). However, the experiment specifically chose handwriting and sketches as the research subjects for one-shot generation tasks.

iii) The core argument proposed by this paper is the generation of samples that resemble human-drawn sketches or handwriting. However, the paper does not provide sufficient elaboration on how to quantify the “human-like” characteristics of the samples, especially with an in-depth analysis from the perspective of stroke features.

**Questions:**

Major:

i)In one-shot drawing tasks, originality constitutes a key evaluation metric. However, when measuring the “human-like” characteristics of samples, the impact of originality is relatively minor, and its role in the evaluation process appears to be more limited.

ii)
Does this paper quantitatively analyze the stroke correlation between the generated sketches and those drawn by humans? Although the generalization curve has been considered in the evaluation of originality and recognizability, the paper does not explicitly reveal the interrelationship between the strokes.

iii)Has this study explored the reasons for or proposed hypotheses about the performance differences exhibited by various inductive bias methods in one-shot drawing tasks?

Minor:

i)Given that classification models may be influenced by their own biases or uneven distributions in the training data, the recognizability of the model does not necessarily equate to the “human-like” level of the samples. Does the paper take this potential issue into consideration?

ii)In the exploration of effective methods to improve the performance of one-shot drawing tasks, did this paper consider approaches other than simply adding the prototype-based and Barlow regularizers with weights?

iii)
In Figure 2, does the “pixel baseline” refer to the use of rasterized sketches during training or testing? On this point, this paper could analyze the differential impact of inductive biases when dealing with vector versus raster sketches.

**Limitations:**

No. How the evaluation criteria effectively align with human judgment or aesthetic standards is a question of considerable research value, especially in generative tasks. Particularly when dealing with sketches or handwriting, the consideration of strokes is an indispensable key dimension that cannot be overlooked.

---

> ### Author Rebuttal · Authors · 2024-08-05
>
> We thank the reviewer ZReU for the detailed comments. Here is our point-by-point answer:
>
> * **About the lack of analysis of the different regularizer components**: We agree with the reviewer that we did not discuss enough the reasons for such differences. As this issue was shared with other reviewers, we have answered this concern in the bullet point 2), in the general rebuttal form.
>
> * **About considering multimodal and more complex datasets**: We acknowledge that the latent diffusion models we use do not face scalability issues and can handle multiple modalities. As our scientific question involves tight comparisons between humans and machines, we focus on tasks accessible for both humans and machines (which is not the case for natural image synthesis.Therefore, the focus on the one-shot drawing task is deliberate and aligned with the scientific question we explore in this article.
>
> * **About the methods used to compare machines and humans and about stroke analysis**: In this article, we used two independent methods to compare humans and machines —the recognizability vs. originality framework and feature importance maps. Both methods lead to the same conclusion: certain regularizers help narrow the gap with humans in one-shot drawing tasks. But we agree with the reviewer that stroke analysis could be another interesting approach to analyze human drawings. We plan to include such analysis in further work, so that we could compare all 3 different comparison methods.
>
> * **About the low originality value on impact when evaluating human samples**: There might be a misunderstanding on this specific point. The originality plays a major role when comparing various types of generative models (e.g. VAEs, GANs, …) to humans as shown by previous authors (see Fig 3 of [1], or in Fig 2 of [2] ). But in this article, we focus on one particular type of generative model that are the latent diffusion models. Because such models tend to fall close to humans in terms of originality, we have rescaled the originality vs. diversity framework to zoom in and better highlight the differences between humans and machines (note the originality scale starting at 0.5). Such a rescaling might be the reason why the reviewer thinks that originality plays a minor role in the evaluation process. It is also important to note that there is an inherent tradeoff between originality and recognizability: while recognizability assesses how likely the data point falls in the classifier decision boundary, originality measures how ‘diffuse’ the sample distribution is (the Fig 1 in [1] well illustrate this trade-off)... Therefore, a very ‘original’ agent (producing highly diverse samples) will tend to have low recognizability as the samples are likely to fall outside of the classifier decision boundary. In Fig 2, we observe that models need to trade their originality for more recognizability to better approximate human performance. We have included two sentences in the main text (line 245) to clear up this misunderstanding.
>
> * **About the possible biases of the classifier to evaluate the recognizability**: We fully agree with the reviewer that the classifier used to evaluate the recognizability might be biased. Hoewer, we think these biases have a low impact on our analysis for two main reasons. First, we use a one-shot classifier to evaluate the recognizability (similar to the original paper that introduces the originality vs recognizability metrics [1]). Such classifiers are less prone to overfitting by construction (as they learn a metric space, see [3] for more explanation), mitigating the impact of potential biases in the training distribution. Second and most importantly, all samples (either human-drawn or machine-generated) are evaluated with the same classifier. Therefore, any potential biases in the recognizability metric will equally affect both human and machine performance, ensuring that the comparison remains meaningful. We propose to include such an explanation in the main text (line 242) to clarify this point.
>
> * **About other approaches to improve performance on the one-shot drawing task** (i.e. more combinations of regularizer): As this specific point was also raised by reviewer vmMb we have run more experiments to systematically explore more combinations of regularizers. Those experiments lead to one more figure. More details are in the bullet point 3 of the general rebuttal form.
>
> * **About the pixel baseline and the rasterization**: Our pixel baseline is indeed a diffusion model (without any latent projection) to learn to generate the distribution of pixel value. It indeed leverages rasterized images, as the original quickdraw images are vectorized.
>
> * **About the alignment of the metrics with human judgment**: We fully agree with the reviewer that having evaluation metrics (such as originality and recognizability) that align with human judgment would make our analysis more impactful. We are currently working on finding more aligned metrics (using harmonization techniques combined with psychophysics experiments). We have added 2 sentences in the discussion section (line 381) to discuss this interesting point. We thank the reviewer for the valuable comment.
>
> We believe that our response, along with the additional curves and paragraph included in the article, addresses most of the reviewer's concerns and should encourage them to raise their rating.
>
> [1] Boutin, Victor, et al. "Diversity vs. Recognizability: Human-like generalization in one-shot generative models." Advances in Neural Information Processing Systems 35 (2022): 20933-20946. \
> [2] Boutin, Victor, et al. "Diffusion models as artists: are we closing the gap between humans and machines?." Proceedings of the 40th International Conference on Machine Learning. 2023. \
> [3] Snell, Jake, Kevin Swersky, and Richard Zemel. "Prototypical networks for few-shot learning." Advances in neural information processing systems 30 (2017).

---

> ### Author Response · Authors · 2024-08-12
>
> Should there be any remaining issues, we are more than willing to engage in further discussion.

---

### Official Review · Reviewer_vmMb · 2024-07-13

**Soundness:** 2
**Presentation:** 2
**Contribution:** 2
**Rating:** 4
**Confidence:** 3

**Summary:**

This paper investigates how different representational inductive biases in Latent Diffusion Models affect their performance on one-shot drawing tasks, aiming to close the gap with human abilities. The authors explore six regularization techniques: KL divergence, vector quantization, classification, prototype-based, SimCLR, and Barlow twins. They evaluate these models using the originality vs. recognizability framework and a novel method for generating feature importance maps. The results show that prototype-based and Barlow regularizations significantly narrow the gap between LDMs and human performance in one-shot drawing tasks. These regularizers outperform standard LDM regularizers (KL and vector quantization) as well as classification and SimCLR regularizers. The authors also demonstrate that the feature importance maps of LDMs with prototype-based and Barlow regularizations align more closely with human attentional strategies. Additionally, they find that combining these two regularizers yields even better results. The study highlights the potential of incorporating specific representational inductive biases in generative models to achieve more human-like generalization capabilities, with implications for both AI advancement and understanding human cognitive processes.

**Strengths:**

- The paper presents a study by systematically exploring various representational inductive biases in Latent Diffusion Models for one-shot drawing tasks. The application of regularization techniques from one-shot classification to generative models provides new insights into improving model performance.
- The authors employ a wide range of regularization techniques and evaluate them using multiple metrics, including the originality vs. recognizability framework and a newly developed method for generating feature importance maps. The statistical analysis of the results adds credibility to their findings.
- This paper demonstrates how specific inductive biases can substantially improve the performance of generative models in one-shot tasks, potentially leading to more versatile and human-like AI systems. It also shows the alignment between the most effective regularizers (prototype-based and Barlow) and prominent neuroscience theories provides interesting insights into human cognitive processes. The paper's findings could have practical applications in areas requiring rapid generalization from limited examples, such as design prototyping or creative tasks.

**Weaknesses:**

- The paper doesn't provide a detailed analysis of how different components of the regularizers contribute to the overall performance. This makes it challenging to understand which specific aspects of each regularizer are most crucial.
- While the authors use human-derived feature importance maps, they don't include a human evaluation of the generated samples. Such an evaluation could provide additional insights into the perceived quality and human-likeness of the generated drawings.
- Although the authors briefly mention combining prototype-based and Barlow regularizers, this aspect is not thoroughly explored. A more systematic investigation of different regularizer combinations could potentially yield even better results.

**Questions:**

- How well do your findings generalize to more complex datasets beyond simple sketches? Have you considered testing your approach on datasets with more detailed or realistic images?

---

> ### Author Rebuttal · Authors · 2024-08-05
>
> We thank the reviewer vmMb for the relevant and valuable comments. Please find below a point-by-point to answer that answer the main concerns of the reviewer :
> * **About the lack of analysis on the impact of the different regularizers on the performance**: In Fig 2, the comparison between the regularized latent diffusion models (shown in color) with the no-regularization baseline (hexagon marker) shows how the different regularizers contribute to the global performance (as measured by the originality vs. recognizability framework). However, we acknowledge the reviewer's concern that we did not discuss enough why the regularizers produce such effects. To mitigate this issue we have added an entire paragraph (at the end of the results section) that better explains these differences. Because it was also requested by another reviewer, we have included this paragraph in the general response (see bullet point 2). We thank the reviewer for raising this issue, as we think it improves the quality of the article.
>
>
> * **About the human evaluation of the samples**: Human evaluation of the generated samples has already been proposed by previous studies. For example, [1] did a Turing test where humans had to distinguish between human-drawn and machine-generated samples. While insightful, such experiments give few insights into how the samples differ from machine to human. On the contrary, we think originality vs. recognizability offers a finer comparison between the distribution of images drawn by humans and those generated by machines.
>
> * **About systematic investigations of the different regularizers combinations**: To address the reviewer’s concern, we have run additional experiments to systematically explore various combinations of regularizations. In particular, we have systematically explored the following combinations of regularization Barlow + Prototype: KL + prototype, SimCLR+Prototype, VQ + Prototype. Among all combinations of regularizers, it is the Barlow + Prototype and the KL + Prototype combinations that perform the best. Interestingly, the good performance of the combination  KL + Prototype was not expected, because the KL regularizer (when used separately) shows a low recognizability (see Fig2). On the other hand, the VQ + Prototype combination shows no improvement compared to the Prototype regularizer. Overall, these additional experiments confirm the potential of combining an unsupervised and a supervised regularizer to match human performance. These experiments lead to one additional figure (included in the one-page PDF allowed in the general response). Note that we have also explored all combinations between unsupervised and the classifier regularizer, but we did not observe any significant improvements (we have included those experiments in the supplementary information of the revised version).
>
> * **About generalization to more complex databases**: Note that the diffusion models as well as the RAEs we use in these articles are all known to scale well to larger datasets [2]. So in theory nothing prevents us from applying the proposed regularization in more complex settings. However, the point of our article is to compare humans with machines. To do so, one needs to leverage tasks that are accessible by both humans and machines, which is not the case with natural image synthesis: humans can hardly produce images that resemble natural images. We have chosen the one-shot drawing task because it offers a leveled playfield to compare humans and machines. To make this point clearer, we have included a sentence in line 133.
>
> Overall, we hope that our detailed response has addressed the reviewer's concerns, and convinced them to increase their rating. Should there be any remaining issues, we are more than willing to engage in further discussion.
>
> [1]: Lake, Brenden M., Ruslan Salakhutdinov, and Joshua B. Tenenbaum. "Human-level concept learning through probabilistic program induction." Science 350.6266 (2015): 1332-1338. \
> [2]: Rombach, Robin, et al. "High-resolution image synthesis with latent diffusion models." Proceedings of the IEEE/CVF conference on computer vision and pattern recognition. 2022.

---

> ### Author Response · Authors · 2024-08-12
>
> Should there be any remaining issues, we are more than willing to engage in further discussion.

---

### Author Rebuttal · Authors · 2024-08-05

We thank the reviewers for their time, their valuable comments and reviews. While the reviewers have acknowledged our “rigorous academic attitude… that strictly adhere to scientific methods” (Reviewer ZReU), as well as the novelty (Reviewers vmMb, YdjR) and significant practical impact of our article (Reviewers vmMb, ZReU, YdjR), we have not yet convinced all reviewers that our article clearly merits acceptance at the NeurIPS conference. Three main concerns are shared among reviewers:  i) our study is conducted on drawings rather than on natural images dataset (Reviewers vmMb, ZReU,  YdjR), ii) we did not discuss enough the effects of each regularizer (Reviewers vmMb, ZReU), and iii) we did not conduct systematic experimental comparisons on the combined effects of the regularizers (Reviewers vmMb, ZReU, YdjR):
1. The one-shot drawing setting (involving sketches dataset) is a deliberate choice. It offers a leveled playfield to compare humans and machines, which is at the heart of our scientific question. Natural image generation is a feat that surpasses human capability, making it not optimal to draw fair comparisons between humans and machines in the one-shot generation setting. We have updated the article to make this point clearer.
2. We fully agree with the reviewers that we did not deepen enough the intuition why some regularizers are better performing than others. To mitigate this issue, we have added an entire paragraph at the end of the results section :

   “The experimental results shown in Fig2 show that not all regularizers are created equal. For the supervised regularizers (Fig 2b) we observe that the prototype regularizer produces more recognizable samples compared to the classification regularizer. This behavior is expected as the classifier learns features separating the categories from the training distribution. However, in the one-shot setting, such features might not be optimal to separate the categories of the (unseen) categories of the testing set [1, 2]. On the other hand, the prototype-based regularizer learns an embedding that clusters samples near their prototypes rather than directly mapping features to labels. Such a strategy is less prone to overfitting and produces more transferable features, making it particularly valuable for few-shot categorization tasks (see [3] for more discussions). Our experiments demonstrate that the prototype regularizer also better generalizes in the one-shot drawing setting. In Fig 2c, we observe that the Barlow regularizer outperforms the SimCLR regularizer in terms of recognizability. We attribute this better performance to the Barlow loss function's ability to disentangle features effectively (as measured by linear probing in [4]). These features also transfer more readily to different datasets ([4]), making the Barlow regularizer a better candidate than the SimCLR regularizer for the one-shot drawing task. Overall, our results demonstrate that the effective representational inductive biases in few-shot learning are also leading to better performance in the one-shot drawing task.”

   In addition, this paragraph also answers the reviewer's ZReU about the ‘lack of clear response to the scientific question’.  Indeed, the new paragraph highlights the fact that regularizers proven effective in the one-shot classification tasks tend to be also effective in the one-shot drawing setting. It therefore gives a clear answer to our scientific question:  “Do representational inductive biases from one-shot classification help narrow the gap with humans in the one-shot drawing task” (clearly stated in line 46).

3. We have included more experiments to systematically study the combined effect of the regularizers. In particular, we have systematically explored the following combinations of regularization Barlow + Prototype: KL + prototype, SimCLR+Prototype, VQ + Prototype. Among all combinations of regularizers, it is the Barlow + Prototype and the KL + Prototype combinations that perform the best. Interestingly, the good performance of the combination  KL + Prototype was not expected, because the KL regularizer (when used separately) shows a low recognizability (see Fig2). On the other hand, the VQ + Prototype combination shows no improvement compared to the Prototype regularizer. Overall, these additional experiments confirm the potential of combining an unsupervised and a supervised regularizer to match human performance. The findings, illustrated by a new figure (see attached pdf) are included in the revised version of the article. This additional analysis requested by the reviewers required the training of around 600 additional latent diffusion models, amounting to about 1800 days/gpu computation time.

We have also included reviewer-specific answers that individually address the reviewer’s concerns. We hope this will convince the reviewers of our article's value and its suitability for publication at NeurIPS.

[1] Snell, Jake, Kevin Swersky, and Richard Zemel. "Prototypical networks for few-shot learning." Advances in neural information processing systems 30 (2017). \
[2] Vinyals, Oriol, et al. "Matching networks for one shot learning." Advances in neural information processing systems 29 (2016).
[3] Li, Xiaoxu, et al. "Deep metric learning for few-shot image classification: A review of recent developments." Pattern Recognition 138 (2023): 109381. \
[4] Zbontar, Jure, et al. "Barlow twins: Self-supervised learning via redundancy reduction." International conference on machine learning. PMLR, 2021.

---

### Comment · Area_Chair_mkhq · 2024-08-12

Dear reviewers,

As the rebuttal period is closing, I am obligated to remind the reviewers to respond to the author rebuttals if possible. As reviewers vmMB, ZReU, and 1Pi8 all lean towards rejection, I especially request these reviewers to respond to the rebuttal before the deadline tomorrow (Aug 13 11:59pm AoE) and to indicate whether the rebuttal addressed any of your concerns.

Gratefully,

AC

---

### Comment · Reviewer_ZReU · 2024-08-13
**Official Comment**

The author's comprehensive response in the rebuttal section effectively addresses most of my initial concerns, particularly by delving into the underlying mechanism of the regularizers' excellent performance. Based on this, I am inclined to give this paper a positive evaluation and change my votes to Borderline Accept.

---

### Decision · Program_Chairs · 2024-09-25

**Decision:**

Accept (poster)

**Comment:**

While use of a combined evaluation of originality and recognizability is not novel in the field, the application of this evaluation framework to human sketches is a non-trivial contribution.  The testing of multiple regularizers, and the addition of combined regularizers, is likely to be impactful for the field of ML models of human-like sketching and character writing.  While the paper could be strengthened by adding evaluation of stroke features or a human evaluation of the human-likeness of the sketches, or a deeper analysis of the performance of the regularizers, I think the paper provides enough of a contribution to be useful to the field as it currently stands.